# Coarse-to-Fine Vision-Language Pre-training with Fusion in the Backbone

**Zi-Yi Dou**[*‡], **Aishwarya Kamath**[*♮], **Zhe Gan**[*†♠], **Pengchuan Zhang**[§], **Jianfeng Wang**[†]
**Linjie Li**[†], **Zicheng Liu**[†], **Ce Liu**[†], **Yann LeCun**[♮], **Nanyun Peng**[‡], **Jianfeng Gao**[†], **Lijuan Wang**[†]

[†]Microsoft   [‡]University of California, Los Angeles   [♮]New York University
{zdou,violetpeng}@cs.ucla.edu, {aish,yann.lecun}@nyu.edu, pengchuanzhang@fb.com
{zhgan,jianfw,linjli,zliu,liuce,jfgao,lijuanw}@microsoft.com

## Abstract

Vision-language (VL) pre-training has recently received considerable attention. However, most existing end-to-end pre-training approaches either only aim to tackle VL tasks such as image-text retrieval, visual question answering (VQA) and image captioning that test high-level understanding of images, or only target region-level understanding for tasks such as phrase grounding and object detection. We present FIBER (**F**usion-**I**n-the-**B**ackbone-based transform**ER**), a new VL model architecture that can seamlessly handle both these types of tasks. Instead of having dedicated transformer layers for fusion after the uni-modal backbones, FIBER pushes multimodal fusion deep into the model by inserting cross-attention into the image and text backbones, bringing gains in terms of memory and performance. In addition, unlike previous work that is either only pre-trained on image-text data or on fine-grained data with box-level annotations, we present a two-stage pre-training strategy that uses both these kinds of data efficiently: (*i*) *coarse*-grained pre-training based on image-text data; followed by (*ii*) *fine*-grained pre-training based on image-text-box data. We conduct comprehensive experiments on a wide range of VL tasks, ranging from VQA, image captioning, and retrieval, to phrase grounding, referring expression comprehension, and object detection. Using deep multimodal fusion coupled with the two-stage pre-training, FIBER provides consistent performance improvements over strong baselines across all tasks, often outperforming methods using magnitudes more data. Code is available at https://github.com/microsoft/FIBER.

## 1 Introduction

Inspired by the success of language model pre-training [11, 51, 42], coupled with the unification of architectures used in the NLP and computer vision communities [12, 4], vision-language pre-training (VLP) [62, 45, 33, 6] has been receiving an increasing amount of attention. It has been proven that VLP can establish state-of-the-art performance on visual question answering [3], visual reasoning [60], image captioning, and image-text retrieval [41]. The pre-training objectives commonly used for these tasks, such as image-text matching, image conditioned masked language modeling and image-text constrastive learning, require multimodal understanding at the image level. Typically, this means the pre-training is done using images at lower resolution (*e.g.*, 384×384), making it possible to scale up training by using large batch sizes.

Recently, it has also been shown that tasks such as image classification and object detection (OD), which have been traditionally viewed as vision-only tasks, can benefit from being cast as VL tasks [50, 25, 34, 26]. Inspired by MDETR [26], GLIP [34] reformulates standard classification-based OD as phrase grounding. This opens up the possibility to leverage VLP for OD, and vice versa,

---

[*]Equal Technical Contribution   [♠]Project Lead   [§]Work done while at Microsoft

36th Conference on Neural Information Processing Systems (NeurIPS 2022).

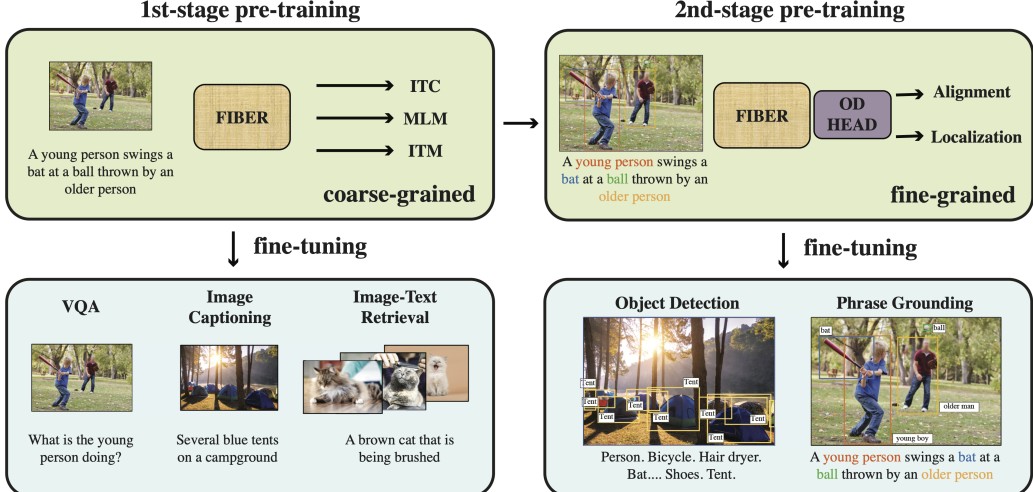

**Figure 1:** The proposed coarse-to-fine pre-training framework for vision-language tasks. We first perform *coarse*-grained pre-training with image-text data for VQA, image captioning and retrieval tasks, and then perform *fine*-grained pre-training with image-text-box data for phrase grounding and object detection tasks. The same FIBER architecture is used for both stages. OD: object detection. MLM: masked language modeling. ITM: image-text matching. ITC: image-text contrastive loss.

and this unification has led to impressive performance on several established OD as well as phrase grounding benchmarks [49]. Since these tasks involve fine-grained image understanding between regions in the image and phrases in the text, and also require prediction of precise bounding boxes at the output, the pre-training typically involves using high resolution input images (*e.g.*, 800×1,333).

Existing multimodal architectures typically do not support both kinds of tasks. Specifically, the fully end-to-end VLP models such as ALBEF [32], METER [13], and SimVLM [67] can achieve the state of the art (SoTA) on image-level understanding tasks, but it is non-trivial to extend them for region-level VL tasks because predicting bounding boxes is typically hard in end-to-end settings. On the other hand, MDETR [26] and GLIP [34] are designed to predict bounding boxes, but have not been shown to support tasks such as image captioning and retrieval. Further, fine-grained pre-training not only requires data with bounding box annotations that are cumbersome to acquire, but the requirement of high input image resolution makes pre-training very costly, especially when using standard Transformer architectures [63] that have quadratic complexity in the size of the image. A natural but challenging question arises: *can we have a unified framework for efficient VL pre-training that benefits both image-level and region-level VL tasks (e.g., both VQA and OD)?*

We answer this question by proposing two ideas: (*i*) a novel model architecture that can handle various types of tasks and pre-training strategies (high and low resolution inputs, image and region level outputs) more efficiently than previous work (see Section 3.1 and 4), and (*ii*) a two-stage pre-training pipeline.

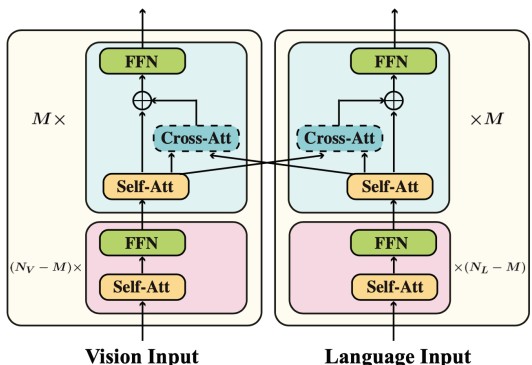

In terms of *architecture*, we present FIBER, shown in Figure 2, which performs deep multimodal fusion in the backbone. Specifically, instead of having a few dedicated transformer layers on top of the image and text encoders for fusion (*e.g.*, as is commonly done in previous work [36, 6, 13, 26, 34]), we propose to directly insert cross-attention modules into the image and text backbones. Additionally, we support the ability to switch between a dual encoder (for fast image retrieval) and a fusion encoder (for VQA and captioning) readily, by switching on or off the cross-attention modules. With the same model architecture, by simply adding an object detection

**Figure 2:** Model architecture for FIBER. Swin transformer is used as the image backbone, simplified here for illustration purposes.

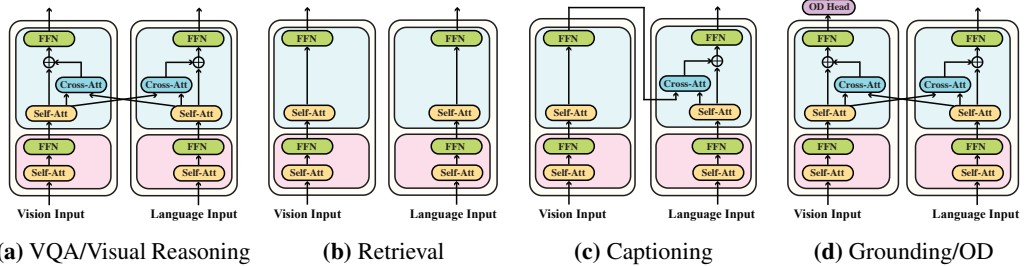

| **(a)** VQA/Visual Reasoning | **(b)** Retrieval | **(c)** Captioning | **(d)** Grounding/OD |

**Figure 3:** FIBER can be readily adapted to various downstream VL tasks, ranging from VQA, image captioning and retrieval, to phrase grounding and object detection (OD).

head (*e.g.*, Dynamic Head [9]) on top, FIBER can be readily extended to visual grounding, referring expression comprehension and (open-vocabulary) OD tasks as well.

By considering the nature of different VL tasks, FIBER is pre-trained with a coarse-to-fine two-stage pipeline, as detailed in Figure 1. Specifically,

- During *coarse*-grained pre-training, FIBER takes low-resolution (384×384) images as input, and is pre-trained with image-text matching, masked language modeling, and image-text contrastive losses, as used in previous work [13, 66, 64]. The pre-trained model can then be directly finetuned for VQA and image captioning tasks (Figure 3a and 3c). By switching off the cross-attention modules, FIBER also automatically functions as a dual encoder for fast image-text retrieval (Figure 3b).

- During *fine*-grained pre-training, FIBER uses the coarse pre-trained model as initialization, in addition to randomly initialized parameters for the OD head. At this stage, the model takes high-resolution (800×1,333) images as input, and is pre-trained with bounding box localization loss and word-region alignment loss, as used in GLIP [34]. We use image-text-box data with ground-truth box annotations for pre-training, and the model can be directly fine-tuned for grounding and detection tasks (Figure 3d).

Compared to fine-grained pre-training, coarse-grained pre-training is easier to scale up, as it only requires paired image-text data which can be easily harvested from the web. Crucially, we show that re-using all the parameters from our coarse-grained pre-trained model for fine-grained pre-training alleviates the requirement for large amounts of box-level annotated data. In our experiments, we show that on fine-grained tasks such as Flickr30k Entities, FIBER using coarse-grained pre-training achieves gains even over previous SoTA (GLIP [34]) that uses 25× more box-level annotated images during the fine-grained pre-training stage. We also show that our architecture is much more efficient in terms of training time on OD tasks, as compared to GLIP .

FIBER is the first end-to-end VLP model that can support VL tasks encompassing image-level and region-level outputs. We conduct experiments on VQAv2 [3], NLVR² [60], COCO captioning [41], NoCaps [1], COCO and Flickr30k image-text retrieval [49], as well as on phrase grounding [49], referring expression comprehension [75], COCO and LVIS detection [17], and a suite of 13 object detection in the wild datasets [34]. We show that our model can provide consistent performance improvement over strong baselines (*e.g.*, METER [13] and GLIP [34]) across tasks.

## 2 Related Work

**VLP for Classical VL Tasks.** ViLBERT [45] and LXMERT [62] were the first two methods to introduce using transformers for VLP. Since then, we have witnessed a boom of VLP methods [33, 30, 59, 73, 22, 71, 81, 38, 7, 35]. Early methods mainly focus on the use of pre-trained object detectors to extract image region features offline, such as UNITER [6], OSCAR [36], VILLA [15] and VinVL [79]. More recently, end-to-end VLP methods that use the image directly as input have become popular. In these approaches, convolution networks or vision transformers [12] are used as the image backbone, with additional transformer layers for modeling multimodal fusion [24, 23, 28, 68, 32, 64]. Prominent examples along this line include ViLT [28], ALBEF [32], SimVLM [67], METER [13], X-VLM [77] and BLIP [31]. These models have achieved the current SoTA on major VL benchmarks such as VQA and image captioning. However, they cannot be directly used for tasks such as object detection.

| Model | VQA[†] | $O(n+m)$ Retrieval[‡] | Captioning | Grounding | OD | End2End |
|---|---|---|---|---|---|---|
| ViLBERT [45], LXMERT [62], UNITER [6] | ✓ | ✗ | ✗ | ✓ | ✗ | ✗ |
| OSCAR [36], VinVL [79] | ✓ | ✗ | ✓ | ✓ | ✗ | ✗ |
| PixelBERT [24], CLIP-ViL [57], ViLT [28] | ✓ | ✗ | ✗ | ✗ | ✗ | ✓ |
| CLIP [50]*, ALIGN [25] | ✗ | ✓ | ✗ | ✗ | ✗ | ✓ |
| VL-T5 [7] | ✓ | ✗ | ✓ | ✗ | ✗ | ✗ |
| METER [13], SimVLM [67] | ✓ | ✗ | ✓ | ✗ | ✗ | ✗ |
| ALBEF [32], FLAVA [58], VLMo [66] | ✓ | ✓ | ✗ | ✗ | ✗ | ✓ |
| BLIP [31], CoCa [74], Flamingo [2] | ✓ | ✓ | ✓ | ✗ | ✗ | ✓ |
| MDETR [26], GLIP [34] | ✓ | ✗ | ✗ | ✓ | ✓ | ✓ |
| UniTAB [70], X-VLM [77], OFA [65] | ✓ | ✗ | ✓ | ✓ | ✗ | ✓ |
| FIBER | ✓ | ✓ | ✓ | ✓ | ✓ | ✓ |

**Table 1: Comparison among different VLP models**. FIBER is the only VLP model that can support all tasks considered. (†) VQA is used as a representative VL classification task. (‡) $O(n+m)$ retrieval denotes model backbones process inputs $O(n+m)$ times given $n$ images and $m$ text sentences during image-text retrieval. (∗) Here, we mainly focus on what tasks CLIP can be directly used for.

**VLP for Vision Tasks.** Recently, it has been shown that image-text data can be used to learn image encoders from scratch [10, 54]. By performing large-scale contrastive pre-training, CLIP [50] and ALIGN [25] display strong zero-shot image classification capabilities. While these models mainly tackle image-level understanding tasks, MDETR [26] extends the end-to-end OD model DETR [4], and uses contrastive learning along with an alignment loss to learn correspondences between image regions and text phrases, opening up the possibility to tackle tasks such as phrase grounding and long-tailed OD using VL models. This has inspired many follow-up works to further enhance the pre-training [37, 72, 46, 69], among which GLIP [34] shows that OD can also be cast as a VL task (*i.e.*, phrase grounding). However, it has not been shown how traditional VL tasks such as VQA, captioning and retrieval can be well supported in GLIP [34] and MDETR [26].

**Unified VL Modeling.** There have been a few recent attempts that try to develop unified VL models. VL-T5 [7] unifies VL tasks as text generation; however, pre-trained object detectors are used for image feature extraction, so the model cannot be end-to-end pre-trained. UniT [20] proposes a multimodal multi-task framework with a unified transformer; however, it can only support VQA and object detection tasks, but not captioning and grounding. GPV [18] proposes a general-purpose vision system, and FLAVA [58] presents a VL system similar to METER [13]; however, they did not evaluate on grounding and detection tasks, and their performance on other VL tasks is still far from SoTA. UniTAB [70] and OFA [65] reformulate grounding as a sequence generation task, by borrowing ideas from Pix2Seq [5]. However, these approaches have not been demonstrated to work on standard OD benchmarks, and also cannot be used as dual encoders for fast image retrieval. Our model is the first work that can support not only VQA, image captioning and $O(n+m)$ retrieval, but also visual grounding and object detection, with impressive performance across all tasks. A detailed comparison is provided in Table 1.

## 3 Method

In this section, we first describe the proposed model architecture in Section 3.1. We then illustrate our two-stage pre-training paradigm in Section 3.2, followed by fine-tuning strategies for all the tasks supported by FIBER in Section 3.3.

### 3.1 Fusion in the Backbone

The architecture of FIBER is shown in Figure 2. Different from models that stack a modality fusion module on top of the vision or language backbones [6, 13], we insert multimodal fusion inside the backbones, and include a gating mechanism for the cross-modal layers (shown in Figure 4). Specifically, at each encoding layer, we have:

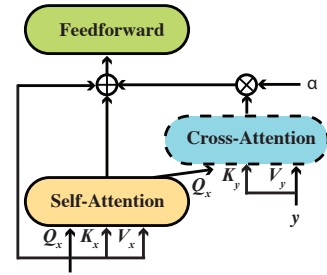

$$\tilde{x} = \text{SELF-ATT}(x),$$
$$x = x + \tilde{x} + \alpha * \text{CROSS-ATT}(\tilde{x}, y), \quad (1)$$
$$x = x + \text{FFN}(x),$$

**Figure 4:** Illustration of performing fusion in the backbone. $(x, y)$ are the (image, text) or (text, image) representations, and $\alpha$ is a learnable scalar.

where $\alpha$ is a learnable parameter initialized to 0. For simplicity, we insert the same number of cross-attention layers into the vision and language backbones.

By inserting cross-attention layers with the gating mechanism, we enable cross-modal interactions without affecting the original computational flow of the backbones at the beginning of model training. Also, we can easily switch off the interactions by setting $\alpha$ to 0, and the backbones can be used in the dual-encoder setting. In addition, compared to stacking a large number of transformer layers on top of the backbones, our approach of inserting cross-attention layers is relatively light-weight and thus more memory-efficient. To illustrate, both GLIP [34] and METER [13] use an additional 110M modality fusion parameters for a base-size model, while FIBER only adds about 26M parameters. During training, the fusion module of FIBER only consumes half of the FLOPs needed by METER (12.35 vs. 24.04 GFLOPs for one instance). We experimented with two other model variants for fusion in the backbone, the details of which are provided in Appendix.

## 3.2 Coarse-to-Fine Pre-training

We divide VL tasks into two categories based on whether or not we need to generate region-level outputs on the image side. While these two kinds of tasks are characteristically different, they both require fusion between the vision and language modalities, and we hypothesize that sharing as many parameters as possible between the model used for these two sets of tasks will be beneficial. Based on this motivation, we propose a two-stage pre-training paradigm, where we first pre-train models with image-level objectives on images at low resolution, and then perform further pre-training with region-level objectives where the input images are at a higher resolution. In this way, the coarse-grained supervision from the first stage can provide good initialization for the second stage for all the shared parameters. FIBER with the same architecture (Swin Transformer [43] and RoBERTa [42]) is used as the backbone for both stages of pre-training.

**Coarse-grained Pre-training.**    For tasks like VQA and captioning, it has been demonstrated [32, 13, 66] that masked language modeling (MLM), image-text matching (ITM), and image-text contrastive (ITC) objectives are helpful for ViT-based VLP models. Following previous work, we use all the three objectives during pre-training. Specifically,

- **For ITC**, the inserted cross-attention modules are switched off, so FIBER functions as a dual encoder. Given a batch of $N$ image-caption pairs, we first compute their representations with our vision and language encoders independently without modality fusion, and then maximize the similarities between $N$ positive image-text pairs while minimizing the similarities between the rest $N^2 - N$ negative pairs, via a contrastive loss.

- **For MLM and ITM**, the inserted cross-attention modules are switched on, so FIBER now functions as a fusion encoder. For MLM, we randomly mask 15% of the input tokens and the model is trained to reconstruct the original tokens. For image-text matching, the model is given an image-text pair and predicts whether they are matched. Following VLMo [66], we sample global hard negatives based on the similarities computed from the above ITC loss.

**Fine-grained Pre-training.**    Most existing VL architectures [6, 62, 26, 31, 65, 7] use vanilla transformers both for encoding the vision as well as language inputs. However, in contrast to tokens in text, the entities of interest in images do not all occur at the same scale. Being able to accurately model the image at different scales is especially important for tasks such as object detection and phrase grounding. To handle this, it is typical in object detection literature to use input images at higher resolutions (800×1333), which becomes problematic when using vanilla transformers that scale quadratically in the length of the input sequence. As mentioned earlier, we use a Swin Transformer [43] as our image encoder, which provides hierarchical representations of the image while having linear complexity in the size of the image. We combine these multi-scale representations using an FPN [39] for object detection training. For fine-grained pre-training, we switch on the cross-attention modules, using FIBER as a fusion encoder. This ensures that the image representations that are passed to the FPN are already text-aware, and is a crucial difference compared to GLIP [34], where the image-text fusion takes place in the object detection head. Once the text-aware image features are extracted by the Swin backbone and image-aware text features are extracted using RoBERTa [42], the image features after the FPN are fed to a DynamicHead [9] which predicts a set of regions. Just as in [34], we compute the dot product between the image region features $R_{\text{TA}}$ and the contextualized

token representations $\boldsymbol{T}_{\text{IA}}$ to compute the grounding score:

$$\boldsymbol{I}_{\text{TA}}, \boldsymbol{T}_{\text{IA}} = \text{FIBER}(I, T), \ \boldsymbol{R}_{\text{TA}} = \text{OD-HEAD}(\boldsymbol{I}_{\text{TA}}), \ S_{\text{GROUNDING}} = \boldsymbol{R}_{\text{TA}} \boldsymbol{T}_{\text{IA}}^{\top}, \qquad (2)$$

where $\boldsymbol{R}_{\text{TA}}$ represents regions that are text aware, produced using the OD-Head that takes as input $\boldsymbol{I}_{\text{TA}}$, which are image representations that are already text-aware and $\boldsymbol{T}_{\text{IA}}$ are the text features that have already attended to the image features. The typical object detection model has a classification head that predicts the label of the object, and a localization head that predicts the bounding box. We follow GLIP [34] by substituting the classification head with the grounding score $S_{\text{GROUNDING}}$. The localization loss is composed of two parts: a centerness loss and GIoU loss, which are used to supervise the box prediction. Taken together, FIBER learns the correspondence between regions in the image and phrases in the text, making it possible to tackle tasks such as phrase grounding and object detection using the same framework. We use ATSS framework [80] in our paper, but our method can be combined easily with other object detectors such as Faster-RCNN [52] and RetinaNet [40] as well.

### 3.3 Adaptation to Downstream Tasks

We now describe how we adapt FIBER to different downstream tasks as depicted in Figure 3.

- **For VL classification tasks such as VQA**, we use FIBER as a fusion encoder. Specifically, the top $M$ layers of the vision and language backbones interact with each other and produce multimodal representations. The final layer representations of the two modalities are concatenated together to generate the final outputs for tasks such as VQA and visual reasoning.

- **For retrieval tasks**, we switch off the inserted cross-attention modules to use FIBER as a dual encoder for fast image-text retrieval.

- **For captioning**, we adapt FIBER by only keeping the image-to-text cross-attentions and using causal masks in the decoding side. The representations of the final image encoding layer are fed into the cross-attention modules. In this way, the model is turned into a seq2seq model [61, 8] and performs captioning in an auto-regressive way.

- **For phrase grounding, object detection and referring expression comprehension**, we use FIBER as a fusion encoder, and the OD-Head introduced during fine-grained pre-training receives image features that are already language aware due to the multimodal representations extracted by FIBER. The pre-trained model is directly used without any modifications for these tasks.

## 4 Experiments

**Pre-training Datasets.** Following previous work [6, 28, 32, 13, 64, 66], we perform coarse-grained pre-training on COCO [41], Conceptual Captions [56], SBU Captions [47], and Visual Genome [29]. The four datasets consist of about 4M images in total. For fine-grained pre-training, we use two data sources: data curated by MDETR [26] after removing the COCO images, and the Objects365 [55] detection dataset, together consisting of about 0.8M images. We ensure that we exclude any data that exists in the validation or test splits of downstream tasks.

| Type of Fusion | COCO Val2017 | GPU-hours V100 (32GB) | Sec/Iter |
|---|---|---|---|
| No Fuse | 53.9 | 511 | 1.31 |
| GLIP-B [34] | 54.6 | 840 | 2.14 |
| FIBER-B | 54.5 | 540 | 1.38 |

**Table 2:** Object detection on COCO [41], without vision-language pre-training. We initialize the text encoder and image backbones using a pre-trained RoBERTa and a Swin transformer pre-trained on ImageNet22k. Our proposed FIBER model achieves the same performance as GLIP [34] while taking much less time to train.

**Architecture.** We adopt Swin-Base [43] and RoBERTa-Base [42] as our vision and text backbones, which are initialized with weights from uni-modal pre-training. We insert cross-attention blocks into the top 6 blocks of the vision and text encoders. The input resolution is $384 \times 384$ for coarse-grained pre-training and $800 \times 1,333$ for fine-grained pre-training. Using a hierarchical vision transformer enables us to efficiently tackle these high resolution tasks, which would be expensive in models such as BLIP [31] that rely on the vanilla transformer architecture. In METER [13], which does explore using a Swin transformer as the image encoder, the multi-modal fusion occurs in layers specifically designed to align the modalities, only after the image and text features are extracted from the uni-modal backbones. This is in contrast to our approach where the hierarchical image features that are used in the FPN for fine-grained training

| Model | #Pretrain Images | VQAv2 | | NLVR$^2$ | | Flickr30k | | COCO | |
|---|---|---|---|---|---|---|---|---|---|
| | | test-dev | test-std | dev | test-P | IR@1 | TR@1 | IR@1 | TR@1 |
| *Base-size models pre-trained on COCO, VG, SBU, and CC datasets* | | | | | | | | | |
| UNITER-B [6] | 4M | 72.70 | 72.91 | 77.18 | 77.85 | 72.5 | 85.9 | 50.3 | 64.4 |
| VILLA-B [15] | 4M | 73.59 | 73.67 | 78.39 | 79.30 | 74.7 | 86.6 | - | - |
| UNIMO-B [35] | 4M | 73.79 | 74.02 | - | - | - | - | - | - |
| ViLT-B [28] | 4M | 71.26 | - | 75.70 | 76.13 | 64.4 | 83.5 | 42.7 | 61.5 |
| ALBEF-B [32] | 4M | 74.54 | 74.70 | 80.24 | 80.50 | 82.8$^\dagger$ | 94.3$^\dagger$ | 56.8$^\dagger$ | 73.1$^\dagger$ |
| VLMo-B [66] | 4M | 76.64 | 76.89 | 82.77 | 83.34 | 79.3 | 92.3 | 57.2 | 74.8 |
| METER-Swin-B [13] | 4M | 76.43 | 76.42 | 82.23 | 83.47 | 79.02 | 92.4 | 54.85 | 72.96 |
| X-VLM [77] | 4M | 78.22 | 78.37 | 84.41 | 84.76 | 86.9$^\dagger$ | 97.0$^\dagger$ | 63.4$^\dagger$ | 81.2$^\dagger$ |
| *Models pre-trained on more data and/or with larger size* | | | | | | | | | |
| VLMo-L [66] | 4M | 79.94 | 79.98 | 85.64 | 86.86 | 84.5 | 95.3 | 60.6 | 78.2 |
| BLIP$_{CapFilt-L}$ [31] | 129M | 78.25 | 78.32 | 82.15 | 82.24 | 87.5$^\dagger$ | 97.2$^\dagger$ | 64.1$^\dagger$ | 81.2$^\dagger$ |
| SimVLM-B [67] | 1.8B | 77.87 | 78.14 | 81.72 | 81.77 | - | - | - | - |
| SimVLM-H [67] | 1.8B | 80.03 | 80.34 | 84.53 | 85.15 | - | - | - | - |
| FIBER-B | 4M | **78.55** | **78.46** | **84.59** | **85.52** | **81.44** | **92.90** | **58.01** | **75.38** |

**Table 3:** Results on VL classification and retrieval. We also include models pre-trained on more data and/or with larger size. FIBER and VLMo use dual encoders for retrieval. (†) ALBEF, X-VLM, and BLIP first use its dual encoder to obtain top-$k$ candidates, and then use its fusion encoder to re-rank the candidates. Our retrieval results with re-ranking are provided in Table 4. All the other models use fusion encoders.

are already language aware, due to the multi-modal fusion being in the backbone. This also lets us avoid adding additional "language-aware deep fusion layers" [34] as part of the OD head as in GLIP, resulting in 1.5x faster training while maintaining performance as shown in Table 2. While in principle it would be possible to use the image features extracted by METER's backbone for object detection, it would be necessary as in GLIP to add additional layers to make the visual features "language-aware" for good detection performance, especially on datasets with limited training data and with rare and infrequent objects.

**Implementation Details.** We perform coarse-grained pre-training for 100k steps with 4,096 batch size on 64 A100 GPUs. We use AdamW [44] with the peak learning rates of 1e-4 for the backbones and 5e-4 for the cross-modal parameters. We use linear warmup over the first 1k steps and linear decay. For fine-grained pre-training, we train for 800k steps on 64 V100 GPUs, with a batch size of 64. We use a learning rate of 1e-5 for the language backbone, and 1e-4 for the rest of the model with a weight decay of 0.01. We use a linear warmup over the first 2k steps and then a constant learning rate, with two learning rate drops by a factor of 10 at 67% and 89% of the total number of steps.

## 4.1 Results on Downstream Tasks

**Vision-Language Classification.** We first experiment on two representative VL classification tasks, including VQAv2 [3] and NLVR$^2$ [60]. As reported in Table 3, we achieve the best performance compared to other models in the same setting. It is worth noting that FIBER pre-trained with 4M images can achieve better performance than BLIP trained with 129M images and SimVLM trained with 1.8B images. The results indicate that introducing fusion modules into the backbone is an effective alternative to appending them on the top of uni-modal backbones.

**Image-Text Retrieval.** In Table 3 we report image retrieval performance in the dual encoder setting, achieving competitive performance on both Flickr30k [49] and COCO [41] retrieval tasks. However, previous work has shown that fusion encoders obtain superior performance, albeit at the cost of efficiency as it involves feeding every image-text pair into the model. To illustrate, on the COCO test data, ranking the similarities between 5K images and 25K captions requires the model to process each image-caption pair 75M times, whereas the dual encoder model only needs 30K forward passes. As shown in Table 4, the fusion encoder can indeed surpass the dual encoder on retrieval tasks by a large margin. In addition, directly ensembling the two models by summing their similarity scores together for each image-caption pair can bring us huge improvements.

| Model | Flickr30k | | | | | | COCO | | | | | |
|---|---|---|---|---|---|---|---|---|---|---|---|---|
| | IR@1 | IR@5 | IR@10 | TR@1 | TR@5 | TR@10 | IR@1 | IR@5 | IR@10 | TR@1 | TR@5 | TR@10 |
| Fiber-ITC | 81.44 | 96.72 | 98.48 | 92.90 | 99.50 | 99.90 | 58.01 | 83.45 | 90.11 | 75.38 | 94.04 | 97.36 |
| Fiber-ITM | 84.10 | 97.54 | 98.88 | 95.10 | 99.60 | 99.90 | 59.03 | 84.04 | 91.03 | 75.14 | 93.88 | 97.36 |
| Fiber-ITC+ITM Ensemble | 90.96 | 98.44 | 99.14 | 96.00 | 99.70 | 100.00 | 69.73 | 90.66 | 94.59 | 80.10 | 95.60 | 97.98 |
| ALBEF [32] | 82.8 | 96.7 | 98.4 | 94.3 | 99.4 | 99.8 | 56.8 | 81.5 | 89.2 | 73.1 | 91.4 | 96.0 |
| X-VLM [77] | 86.1 | 97.4 | 98.7 | **96.8** | **99.8** | 100.0 | 63.1 | 85.7 | 91.6 | **80.4** | 95.5 | **98.2** |
| Fiber-Rerank-10 | 90.94 | 98.16 | 98.48 | 95.80 | 99.60 | 99.90 | 68.71 | 87.69 | 90.09 | 79.66 | 95.34 | 97.36 |
| Fiber-Rerank-20 | 90.10 | 98.38 | 99.14 | 95.90 | **99.80** | 100.00 | 69.32 | 89.52 | 93.33 | 79.78 | 95.20 | 97.66 |
| Fiber-Rerank-50 | **91.08** | 98.50 | **99.37** | 96.10 | 100.00 | 100.00 | 69.58 | 90.41 | 94.35 | 79.98 | 95.40 | 97.76 |
| Fiber-Rerank-100 | 91.02 | **98.54** | 99.34 | 96.00 | 99.70 | 100.00 | **69.63** | 90.54 | 94.47 | 80.06 | **95.60** | 97.96 |

**Table 4:** Additional results on image-text retrieval, where ($i$) the fusion encoder is used for retrieval, or ($ii$) the dual encoder is first used to obtain top-$k$ candidates, and then the fusion encoder is used to re-rank the candidates. We also provide a full set of results on all evaluation metrics.

| Model | #Pretrain Images | COCO | | | | NoCaps Val | | NoCaps Test | |
|---|---|---|---|---|---|---|---|---|---|
| | | B@4 | M | C | S | C | S | C | S |
| *Models trained without CIDEr optimization* | | | | | | | | | |
| UFO-B [64] | 4M | 36.0 | 28.9 | 122.8 | 22.2 | 80.7 | 12.5 | 78.8 | 12.5 |
| ViTCAP [14] | 4M | 36.3 | 29.3 | 125.2 | 22.6 | - | - | - | - |
| METER-CLIP-B [13] | 4M | 38.8 | 30.0 | 128.2 | 23.0 | - | - | - | - |
| X-VLM [77] | 4M | 39.8 | - | 133.1 | - | - | - | - | - |
| VinVL-B [79] | 5.7M | 38.2 | 30.3 | 129.3 | **23.6** | - | - | - | - |
| BLIP_CapFilt-L [31] | 129M | 39.7 | - | 133.3 | - | 109.6 | 14.7 | - | - |
| LEMON-B [21] | 200M | 40.3 | 30.2 | 133.3 | 23.3 | 106.8 | 14.1 | - | - |
| SimVLM-B [67] | 1.8B | 39.0 | 32.9 | 134.8 | 24.0 | - | - | 94.8 | 13.1 |
| Fiber-B | 4M | 39.1 | 30.4 | 128.4 | 23.1 | 88.6 | 13.0 | 86.0 | 12.9 |
| Fiber-GOLD-B | 4M | **40.3** | **30.7** | **133.6** | **23.6** | **92.8** | **13.4** | **90.6** | **13.4** |
| *Models trained with CIDEr optimization* | | | | | | | | | |
| ViTCAP [14] | 4M | 41.2 | 30.1 | 138.1 | 24.1 | 89.2 | 12.7 | - | - |
| X-VLM [77] | 4M | 41.3 | - | 140.8 | - | - | - | - | - |
| VinVL-B [79] | 5.7M | 40.9 | 30.9 | 140.4 | **25.1** | 94.3* | 13.1* | 92.5* | 13.1* |
| LEMON-B [21] | 200M | 41.6 | 31.0 | 142.7 | 25.1 | - | - | - | - |
| Fiber-B | 4M | 42.8 | 31.0 | 142.8 | 24.3 | 96.7 | 13.4 | 94.1 | 13.4 |
| Fiber-GOLD-B | 4M | **43.4** | **31.3** | **144.4** | 24.6 | **99.2** | **13.7** | **97.1** | **13.8** |

**Table 5:** Results of base-size models on image captioning. We grey models pre-trained on larger magnitudes of data. Numbers with '*' are obtained with constrained beam search during inference and without VLP. The complete results on all metrics are provided in Appendix. B@4: BLEU@4, M: METEOR, C: CIDEr, S: SPICE.

Further, we explore combining the strengths of both strategies by performing re-ranking as in [16, 31, 32]. Specifically, we first retrieve the top-$k$ most similar instances using the dual encoder setup, and then add the similarity scores between the given instance and the top-$k$ candidates provided by the fusion encoder to the original scores to perform retrieval. From Table 4, we can see that this strategy provides a balance between efficiency and performance, and that just re-ranking the top-10 instances can achieve comparable performance with ensembling.

**Image Captioning.** We also evaluate our models on COCO [41] and NoCaps [1] captioning to test whether FIBER can be adapted to generation tasks. As in Table 5, FIBER can achieve better performance than models trained on the same data with and without CIDEr optimization [53]. We find that integrating GOLD [48] into FIBER can bring significant improvements, outperforming models trained with hundreds of millions of images. Notably, we establish the absolute state-of-the-art CIDEr scores on COCO for base-size models. Considering that FIBER is not pre-trained to perform captioning, the results demonstrate the strong generalization ability of FIBER.

**Phrase Grounding.** Our fine-grained pre-training stage incorporates Flickr30k entities grounding data, and we achieve 87.4 on the Recall@1 metric on the test set without any subsequent fine-tuning. This not only surpasses the current SoTA [34] using a smaller sized model (Swin-B compared to their Swin-L), but also uses 25x less fine-grained data. Our FIBER model is able to leverage the image-text coarse-grained pre-training stage better, instead of relying on expensive pseudo-labelling of large

| Model | Image Backbone | #Pretrain Images (fine-grained) | Flickr30k Val | | | Flickr30k Test | | |
|---|---|---|---|---|---|---|---|---|
| | | | R@1 | R@5 | R@10 | R@1 | R@5 | R@10 |
| Visual-BERT [33] | ResNet-101 | 120k | 70.4 | 84.5 | 86.3 | 71.3 | 85.0 | 86.5 |
| MDETR [26] | EN-B5 | 200k | 83.6 | 93.4 | 95.1 | 84.3 | 93.9 | 95.8 |
| GLIP [34] | Swin-B | 860k | 85.7 | 95.0 | 96.2 | 86.1 | 95.5 | 96.4 |
| *Models pre-trained on more data and/or with larger size* | | | | | | | | |
| GLIP [34] | Swin-L | 27M | 86.7 | 96.4 | 97.9 | 87.1 | 96.9 | 98.1 |
| FIBER-B | Swin-B | 860k | **87.1** | **96.1** | 97.4 | **87.4** | **96.4** | 97.6 |
| w/o C.G. VLP | Swin-B | 860k | 86.2 | 96.0 | **97.6** | 86.5 | **96.4** | **97.7** |

**Table 6:** Phrase grounding performance on Flickr30k entities dataset. We reproduce GLIP-Base sized results, and GLIP-Large sized results are taken from [34]. FIBER with Base size outperforms a GLIP-L which is trained with 25x more fine-grained data on the R@1 metric. Further, FIBER without coarse-grained VL pretraining outperforms GLIP-B when trained on the same fine-grained data.

| Model | Pre-training data | | RefCOCO | | | RefCOCO+ | | | RefCOCOg | |
|---|---|---|---|---|---|---|---|---|---|---|
| | Im-Txt | Im-Txt-Box | val | testA | testB | val | testA | testB | val | test |
| MDETR-B [26] | | ✓ | 87.51 | 90.40 | 82.67 | 81.13 | 85.52 | 72.96 | 83.35 | 83.31 |
| UNICORN-B [70] | | ✓ | 88.29 | 90.42 | 83.06 | 80.30 | 85.05 | 71.88 | 83.44 | 83.93 |
| *Models pre-trained on more data and/or with larger size* | | | | | | | | | | |
| UNITER-L [6] | ✓ | | 81.41 | 87.04 | 74.17 | 75.90 | 81.45 | 66.70 | 74.86 | 75.77 |
| VILLA-L [15] | ✓ | | 82.39 | 87.48 | 74.84 | 76.17 | 81.54 | 66.84 | 76.18 | 76.71 |
| OFA-L [65] | ✓ | ✓ | 90.05 | 92.93 | 85.26 | 84.49 | 90.10 | 77.77 | 84.54 | 85.20 |
| FIBER-B | ✓ | ✓ | **90.68** | **92.59** | **87.26** | **85.74** | **90.13** | **79.38** | **87.11** | **87.32** |

**Table 7:** Results on referring expression comprehension datasets.

web-scale corpus and subsequent high-resolution training on this generated fine-grained data as in [34]. We also compare our approach without using any coarse-grained VL training (image encoder initialized to Swin-B weights from IN22k, and text encoder initialized to pre-trained RoBERTa), and even in this setting, we are able to outperform a similarly sized GLIP model (GLIP-B), proving that our fusion in the backbone is better at capturing fine-grained image-text understanding.

**Referring Expression Comprehension (REC).** In contrast to many previous works [6, 15, 45] that tackle the REC task by re-ranking object proposals provided by an off-the-shelf detector, we follow [26] to directly predict the bounding box for the given referring expression. Using our proposed two stage pre-training, FIBER achieves better performance than current SoTA [65] that uses a Large sized model. Notably, on RefCOCOg [76], which contains much longer referring expressions than in Ref-COCO/RefCOCO+ [27], we observe more than 2 points boost over OFA-L. On the challenging testB split of both RefCOCO and RefCOCO+, FIBER outperforms current SoTA, OFA-L.

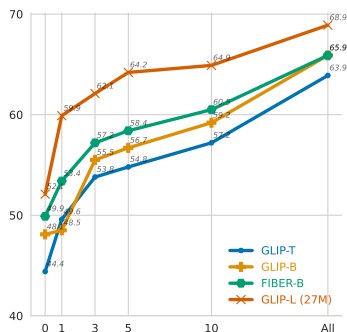

**Figure 5:** Few-shot results on the aggregated 13 ODinW datasets.

**Object Detection.** We report FIBER results on two standard object detection benchmarks, COCO [41] and LVIS [17], in zero-shot transfer[1] as well as fine-tuned settings in Table 8. The LVIS dataset consists of a long-tail of object classes, and is a popular test-bed for evaluating models on their generalization capabilities and robustness to class imbalance. On the APr metric, which is the Average Precision on

---
[1]Following [50, 78], we consider zero-shot transfer to mean that during pre-training we may have seen relevant data but it is not used for training for the task of interest. For instance, our coarse-grained pre-training includes some images from COCO (without any box information), but we do not have any COCO images in our fine-grained training that we use to train the object detection head.

| Model | COCO Val 2017 | LVIS MiniVal | | | | ODinW |
|---|---|---|---|---|---|---|
| | AP | APr | APc | APf | AP | |
| | Zero-shot/Fine-tune | Zero-shot/Fine-tune | | | | Zero-shot/Fine-tune |
| Mask R-CNN [19] | - | - /26.3 | - /34.0 | - /33.9 | - /33.3 | - |
| MDETR [26] | - | - /20.9 | - /24.9 | - /24.3 | - /24.2 | - |
| GLIP-T [34] | 46.7/55.1 | 17.7/ - | 19.5/ - | 31.0/ - | 24.9/ - | 44.4/63.9 |
| GLIP-B [34] | 48.1/57.0 | 17.0/31.3 | 23.9/48.3 | 35.9/56.9 | 29.1/51.0 | 44.8/65.8 |
| *Models pre-trained on more data and/or with larger size* | | | | | | |
| GLIP-L [34] | 49.8/60.8 | 28.2/ - | 34.3/ - | 41.5/ - | 37.3/ - | 52.1/68.9 |
| FIBER-B | **49.3/58.4** | **29.5/50.0** | **32.2/56.9** | **40.1/58.1** | **35.8/56.9** | **47.0/65.9** |

**Table 8:** Zero-shot transfer and fine-tuning results for object detection on COCO, LVIS and the average over 13 datasets for object detection in the wild. Detailed scores on the 13 datasets are presented in the Appendix. FIBER achieves better AP across the board compared to similarly sized GLIP-B, trained on the same amount of fine-grained data. On rare objects in LVIS, FIBER outperforms GLIP-L trained on 25x more fine-grained data. Results without coarse-grained pre-training are provided in the Appendix.

rare objects, FIBER outperforms GLIP-L which is a bigger model and also trained with $25\times$ more fine-grained data.

We also report zero-shot and fine-tuned results on a suite of 13 ODinW (object detection in the wild) datasets, spanning various domains and show consistent performance improvements over previous SoTA. Additionally, in Figure 5, we report few-shot results aggregated across these 13 datasets and show better data efficiency over GLIP-B trained with the same fine-grained data.

**Ablation Study.** In Appendix **??** and **??**, we have provided detailed ablations that guided our architecture design, including ablations on fusion strategies, pre-training objectives, architecture for captioning, and additional results on open-ended VQA, and detailed few-shot ODinW results. Due to the space limit, these ablations and additional results are only provided in the Appendix. Some important observations are summarized below. $(i)$ Co-attention works similarly to merged attention for fusion in the backbone. $(ii)$ Adding a gating parameter in co-attention allows the addition of fusion in more layers, and also gives better performance than merged attention. $(iii)$ Adding co-attention in the last 6 layers provides a balance between performance and efficiency. $(iv)$ MLM, ITM with hard negative mining, and ITC are all important pre-training objectives for training FIBER-style models.

## 5   Conclusion

We propose $(i)$ FIBER, a novel architecture and $(ii)$ a coarse-to-fine pre-training pipeline. We perform extensive experiments and show consistent improvements over strong baselines across a diverse set of tasks. The results demonstrate the effectiveness of FIBER coupled with our pre-training strategy, by setting new SoTA scores while at the same time reducing the requirement of expensive box-level annotations. Future directions include scaling our models and extending our framework to other modalities.

The approach introduced in our work can potentially inherit undesirable societal biases that exist in our pre-training data. Careful debiasing and filtering of data should be undertaken before real-life deployment of our work. Additionally, pre-training can induce environmental costs, and minimizing these costs is an avenue that we plan to explore further.

## Acknowledgement

We would like to thank Nguyen Bach, Jiayuan Huang, and Luis Vargas for their support. We also thank Wenhui Wang, Li Dong, Furu Wei, Bin Xiao, and Lu Yuan for their helpful discussions. We also thank Liunian Harold Li and Te-Lin Wu for their feedback on the manuscript. Aishwarya is supported in part by the National Science Foundation under NSF Award 1922658. Zi-Yi is supported in part by the DARPA Machine Common Sense (MCS) program under Cooperative Agreement N66001-19-2-4032 and NIH R01HL152270.

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
