# A  Appendix

## A.1  Implementation Details

**Vision-Language Classification.**    For the VL classification tasks, we follow METER [5] to set the hyper-parameters. Specifically, we fine-tune our models with the peak learning rates of 2e-5 for the backbones, 1e-4 for the cross-modal parameters, and 1e-3 for the head layer for 10 epochs. The batch size is set to 512. The image resolutions are set to 576 for VQAv2 and 384 for $NLVR^2$. We evaluate models with the VQA scores for VQAv2 and accuracy for $NLVR^2$. RandAugment [4] is used during the downstream fine-tuning stage.

**Image-Text Retrieval.**    For image-text retrieval, we remove the cross-attention layers in the backbones and use the dual encoder architecture. We set the peak learning rates to 2e-5 for the backbones and 1e-4 for the head layer. The batch size is set to 1024. The image resolutions are set to 576 for both COCO and Flickr30k. We evaluate on the Recall@1,5,10 metrics for both text and image retrieval.

**Image Captioning.**    For image captioning, we only keep the image-to-text attentions and feed the image representations in the last layer of the image encoder to the cross-attention modules. In this way, the model is turned into a standard seq2seq model, and we use the causal mask in the decoding side and predict outputs auto-regressively. We first train our models with the cross-entropy loss for 5 epochs with the peak learning rates of 5e-5 for the backbones, and 2.5e-4 for the rest of the parameters. Then, we fine-tune it with GOLD [9] for 5 epochs as it is efficient and has proven to be effective when the model input can correspond to different outputs. We set the peak learning rate to 1e-5 for the backbones during GOLD training. For CIDEr optimization, the learning rate is further reduced to 1e-6 and we train the models for 3 epochs. The batch size is set to 512. We use a beam size of 5 during inference and do not use constrained beam search. We use the same model when testing on COCO and NoCaps, and we evaluate on BLEU [10], METEOR [2], CIDEr [12], and SPICE [1] metrics.

**Phrase Grounding.**    For phrase grounding on Flickr30k, we do not further fine-tune the model after fine-grained pre-training, and just directly evaluate on the Recall@ 1,5,10 metrics.

**Referring Expression Comprehension (REC).**    For the REC datasets, we use batch size 16 and fine-tune on the respective dataset for 20 epochs. We use a warmup of 2000 steps, with a peak learning rate of 1e-5 for both the OD head as well as the rest of the model's paramaters, with two learning rate drops at 67% and 89% of the total number of steps. We switch off the horizontal flip augmentation during REC training, as we find that it adversely affects the performance, especially on the RefCOCO dataset, which includes many examples having degenerate language such as just "left" or "right" rather than using descriptive words for the referring expressions.

**Object Detection.**    For both COCO and LVIS detection, we train for 24 epochs, with batch size 32, with a learning rate of 1e-5 for the whole model, with two learning rate drops at 67% and 89% of the total number of steps. For the ODinW datasets, we fine-tune for 12 epochs, with early stopping based on the validation accuracy.

The object detection data is constructed as follows - The object category names are directly used in their text form separated by full stops as input to the text encoder. We follow the same protocol as in GLIP [3] to be comparable to their experiments. More specifically, the input text will look like this: "person. bicycle. car. .... toothbrush", and the model will learn how to ground image regions into these object names. An example of input and output predicted by the model can be seen in Fig. 4.

## A.2  Ablation Study

**Ablation Study on the Fusion Strategies.**    We perform ablation studies on our fusion module. We investigate three different fusion strategies as shown in Figure 1. Merged attention concatenates representations from the two input modalities and feeds them into the self-attention layer for fusion. Note that here the key and value matrices for the two modalities are different. On the other hand, co-attention inserts a cross-attention layer into each of the encoding layer. The insertion of the

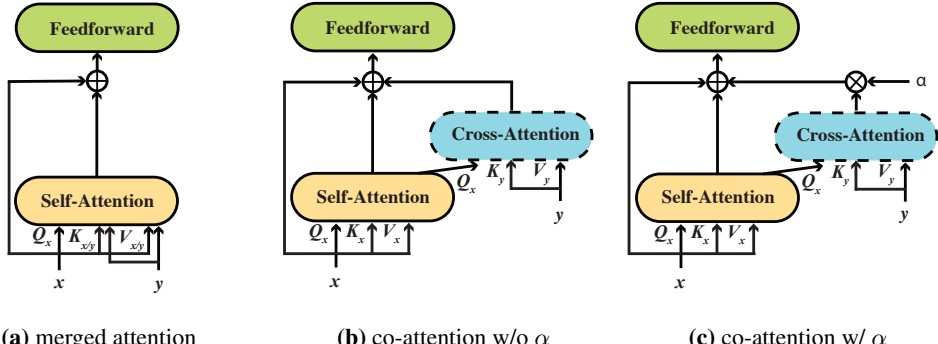

| | | |
|:---:|:---:|:---:|
| **(a)** merged attention | **(b)** co-attention w/o $\alpha$ | **(c)** co-attention w/ $\alpha$ |

**Figure 1:** Different strategies for fusion in the backbone. $(\boldsymbol{x}, \boldsymbol{y})$ are the (image, text) or (text, image) representations, and $\alpha$ is a learnable scalar.

| Model | #Fusion Layers | #Fusion Params. | VQAv2 |
|:---:|:---:|:---:|:---:|
| No Fusion | 0 | 0M | 65.65 |
| Merged Attention | 3 | 7.9M | 71.24 |
| | 6 | 12.6M | 70.67 |
| Co-attention w/o $\alpha$ | 3 | 16.1M | 70.84 |
| | 6 | 26.0M | 68.13 |
| Co-attention w/ $\alpha$ | 3 | 16.1M | 71.20 |
| | 6 | 26.0M | 71.97 |
| | 9 | 35.8M | 72.10 |
| | 12 | 45.6M | 72.08 |

**Table 1:** Ablation study on the fusion strategies. Results are obtained by directly fine-tuning models initialized with uni-modally pre-trained parameters and without VLP. Results on VQAv2 are on test-dev set.

| Pre-training Objectives | | | | VQAv2 | Flickr30k | |
|:---:|:---:|:---:|:---:|:---:|:---:|:---:|
| MLM | ITM | ITM-hard | ITC | test-dev | IR@1 | TR@1 |
| ✓ | ✓ | ✗ | ✗ | 72.47 | 65.50 | 79.30 |
| ✓ | ✗ | ✗ | ✓ | 74.16 | 73.74 | 87.70 |
| ✗ | ✗ | ✓ | ✓ | 67.45 | 75.20 | 87.00 |
| ✓ | ✓ | ✗ | ✓ | 74.49 | 73.58 | 87.80 |
| ✓ | ✗ | ✓ | ✓ | 75.98 | 75.26 | 87.50 |

**Table 2:** Ablation study on the pre-training objectives and whether the hard negative mining strategy is necessary in the coarse-grained pre-training stage.

cross-attention layer offers the flexibility of controlling to what extent we want the two modalities to fuse together as we can easily introduce an $\alpha$ term into the module as in Figure **??**.

As shown in Table 1, we compare the three fusion strategies by directly fine-tuning our models without performing VLP for efficiency. We use Swin Transformer and RoBERTa as our vision and text backbones and load their pre-trained parameters for initialization. We set the image resolution to 224×224. We can see that merged attention and co-attention achieve comparable performance without $\alpha$. For both strategies, increasing the number of fusion layers can lead to performance drop. However, after introducing $\alpha$, we can see significant improvements of co-attention, indicating the importance of having an explicit controlling/gating mechanism for fusion in the backbone.

After the $\alpha$ term is introduced, we can increase the number of fusion layers and achieve robust performance. Based on the ablation results, we choose to fuse the top 6 layers of the backbones as it can achieve a good accuracy-efficiency trade-off.

**Ablation Study on Pre-training Objectives.** Following previous work [7, 5, 14], we pre-train our models with image conditioned masked language modeling, image-text matching with hard negative mining, and image-text contrastive losses during the coarse-grained pre-training stage. In this part,

| Model | OD on COCO | OD on LVIS | ODinW | RefCOCO+ |
|---|---|---|---|---|
| | Zero-shot/Fine-tune | Zero-shot/Fine-tune | Zero-shot | Val/TestA/TestB |
| OFA-L | - | - | - | 84.49/90.10/ 77.77 |
| GLIP-B | 48.1/57.0 | 29.1/51.0 | 44.8 | - |
| FIBER-B w/o C.G. VLP | 48.9/57.8 | 31.6/55.8 | 45.1 | 85.04/88.82/78.59 |
| FIBER-B | 49.3/58.4 | 35.8/56.9 | 47.0 | 85.74/90.13/79.38 |

**Table 3:** Ablation study on our proposed two-stage pre-training strategy.

| Vision Encoder | Text Encoder | VQAv2 |
|---|---|---|
| Swin | RoBERTa | 71.97 |
| Swin | BERT | 71.86 |
| CLIP-ViT | RoBERTa | 71.37 |

**Table 4:** Results of different vision and text backbones for FIBER without VLP.

we ablate each of the pre-training objectives and evaluate our models on both VQAv2 and Flickr30k retrieval tasks. Specifically, we use Swin Transformer and RoBERTa as our vision and text backbones and load their pre-trained parameters for initialization. The image resolution is set to $224 \times 224$ and we pre-train models for 100k steps with 1,024 batch size. We use AdamW with the peak learning rates of 1e-4 for the backbones and 5e-4 for the cross-modal parameters. We use linear warmup over the first 1k steps and linear decay.

As shown in Table 2, we can see that removing any of the pre-training objectives can lead to performance drop, and hard negative mining can bring improvements on both VQA and retrieval tasks. Masked language modeling is most effective for VQA, while removing it will not hurt the retrieval performance. This set of experiments demonstrates that all of the objectives are necessary for our models to obtain good performance.

**Ablation Study on the Two-Stage Pre-training.** In this paper, we propose a coarse-to-fine pre-training strategy for handling VL tasks of different kinds. In this paragraph, we remove the coarse-grained pre-training stage and only pre-train the models with image-text-box data and see how it performs. As shown in Table 3, we see gains across both tasks when utilizing the coarse-grained pre-training. Similar to the case of Flickr30k, on RefCOCO+ the coarse-grained pre-training helps FIBER to get better performance than large-sized model trained with more data. In addition, note that without the coarse-grained pre-training, the only difference between FIBER and GLIP is the architectural difference, and the fact that FIBER can still outperform GLIP in this setting demonstrates the effectiveness of our proposed architecture.

**Ablation Study on Different Backbones.** While previous work [5] has compared different vision and text backbones for VLP models, we investigate if their conclusions stil apply in our settings. Specifically, we try BERT and RoBERTa for our text encoder and CLIP-ViT and Swin Transformer for our image encoder. As shown in Table 4, we can see that RoBERTa and Swin Transformer perform slightly better than BERT and CLIP-ViT before VLP, which is consistent with previous findings in METER [5]. Note that while CLIP-ViT has the potential to perform better than Swin Transformer after VLP, it is hard to be adapted for region-level tasks such as object detection. Therefore, pairing Swin Transformer with RoBERTa is the optimal configuration in our settings.

### A.3   Additional Results

**Additional Results on Image Captioning.** For image captioning, we evaluate the models with BLEU-4, METEOR, ROUGE-L, CIDEr, and SPICE metrics on COCO and NoCaps. On NoCaps, we have fine-grained evaluation results on different domains, including in-domain, near-domain, out-domain, and entire domain settings. In this part, we provide the complete evaluation results in Table 5, 6 and 7. We can see that both GOLD and CIDEr optimization can improve the model performance across metrics. We also see a noticeable performance drop when evaluating our models on out-of-domain data, but complementary methods such as constrained beam search can be used to

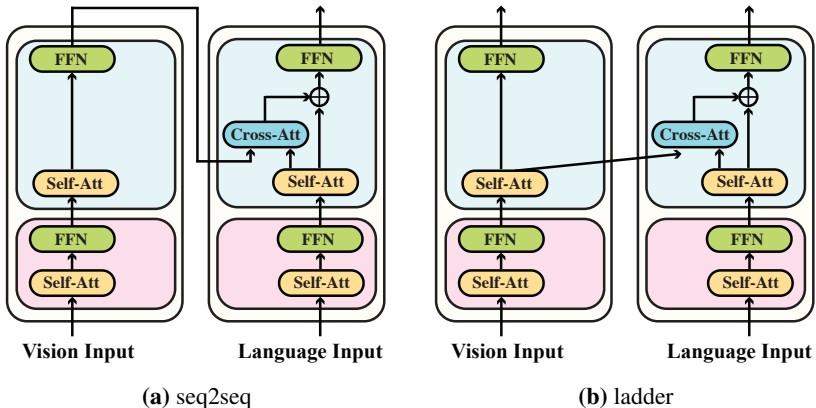

| | (a) seq2seq | | (b) ladder |
|---|---|---|---|

**Figure 2:** When adapting FIBER to image captioning, we can either use the seq2seq structure or the ladder architecture as in pre-training.

| Model | COCO | | | | |
|---|---|---|---|---|---|
| | BLEU@4 | METEOR | ROUGE-L | CIDEr | SPICE |
| *Models trained without CIDEr optimization* | | | | | |
| FIBER-Ladder-B | 38.6 | 30.1 | 58.8 | 127.5 | 22.8 |
| FIBER-B | 39.1 | 30.4 | 59.3 | 128.4 | 23.1 |
| FIBER-GOLD-B | 40.3 | 30.7 | 60.0 | 133.6 | 23.6 |
| *Models trained with CIDEr optimization* | | | | | |
| FIBER-B | 42.8 | 31.0 | 61.5 | 142.8 | 24.3 |
| FIBER-GOLD-B | 43.4 | 31.3 | 61.8 | 144.4 | 24.6 |

**Table 5:** The complete set of results on COCO image captioning, with another model variant FIBER-Ladder. See Figure 2 for details.

alleviate the issue. Also, training our models with more captioning data should also be helpful in these settings.

Also, in the main paper, we adapt FIBER for image captioning by turning it into a standard seq2seq model as in Figure 2a, where the output of the final encoding layer will be fed into the image-to-text cross-attention modules. Another possible design is to keep the ladder structure as we used in pre-training (Figure 2b), so there can be less mismatching between pre-training and fine-tuning. As shown in Table 5, the two architectures can achieve comparable performance. Considering that the seq2seq architecture is more widely adopted in the current literature, we decide to use the seq2seq architecture for image captioning.

**Open-ended VQA.** In most existing literature, VQA is treated as a classification task, where a vocabulary of some most frequent answers are constructed and VL models predict which answer corresponds to the given question based the constructed vocabulary. However, question answering is inherently open-ended. Since we can turn our models into a generative model by fine-tuning on image captioning, we also investigate if our models can perform open-ended VQA in this part.

Following [3], we break down VQA questions into in-domain and out-of-domain questions,

| Model | Open-ended VQA | | |
|---|---|---|---|
| | In-D | Out-D | overall |
| VL-T5 [3] | 71.4 | 13.1 | 67.9 |
| VL-BART [3] | 72.1 | 13.2 | 68.6 |
| SimVLM-B [15] | 78.3 | 25.8 | 75.2 |
| FIBER-B | 75.9 | 14.7 | 71.6 |

**Table 8:** Results on open-ended VQA. We follow [3] to split the data into in domain (In-D) and out of domain (out-D).

| Model | in-domain | | | | | near-domain | | | | | out-domain | | | | | entire | | | | |
|---|---|---|---|---|---|---|---|---|---|---|---|---|---|---|---|---|---|---|---|---|
| | B@4 | M | R | C | S | B@4 | M | R | C | S | B@4 | M | R | C | S | B@4 | M | R | C | S |
| *Models trained without CIDEr optimization* | | | | | | | | | | | | | | | | | | | | |
| FIBER-B | 29.7 | 30.0 | 58.2 | 98.5 | 13.9 | 24.4 | 27.5 | 55.6 | 88.2 | 13.0 | 18.0 | 25.4 | 53.5 | 82.8 | 12.2 | 23.9 | 27.5 | 55.6 | 88.6 | 13.0 |
| FIBER-GOLD-B | 29.7 | 30.1 | 58.2 | 100.6 | 14.0 | 26.8 | 28.2 | 57.0 | 92.9 | 13.5 | 18.3 | 25.8 | 54.3 | 86.6 | 12.8 | 25.5 | 28.0 | 56.6 | 92.8 | 13.4 |
| *Models trained with CIDEr optimization* | | | | | | | | | | | | | | | | | | | | |
| FIBER-B | 34.2 | 30.9 | 60.0 | 108.9 | 14.0 | 28.8 | 28.4 | 58.2 | 96.0 | 13.5 | 19.8 | 26.0 | 55.6 | 90.1 | 12.7 | 27.7 | 28.3 | 57.9 | 96.7 | 13.4 |
| FIBER-GOLD-B | 35.4 | 31.2 | 60.6 | 110.3 | 14.3 | 30.5 | 29.0 | 58.9 | 99.5 | 13.8 | 20.4 | 26.0 | 55.6 | 90.2 | 12.8 | 29.1 | 28.7 | 58.5 | 99.2 | 13.7 |

**Table 6:** The complete set of results on the NoCaps validation set. B@4: BLEU@4, M: METEOR, R: ROUGE-L, C: CIDEr, S: SPICE.

| Model | in-domain | | | | | near-domain | | | | | out-domain | | | | | entire | | | | |
|---|---|---|---|---|---|---|---|---|---|---|---|---|---|---|---|---|---|---|---|---|
| | B@4 | M | R | C | S | B@4 | M | R | C | S | B@4 | M | R | C | S | B@4 | M | R | C | S |
| *Models trained without CIDEr optimization* | | | | | | | | | | | | | | | | | | | | |
| FIBER-B | 28.6 | 29.5 | 57.7 | 92.8 | 13.6 | 25.6 | 28.0 | 56.1 | 87.3 | 13.0 | 16.2 | 24.4 | 52.1 | 76.4 | 11.6 | 24.3 | 27.6 | 55.6 | 86.0 | 12.9 |
| FIBER-GOLD-B | 29.9 | 30.1 | 58.4 | 95.9 | 14.1 | 28.0 | 28.7 | 57.4 | 92.0 | 13.5 | 18.4 | 25.3 | 53.4 | 81.0 | 12.3 | 26.5 | 28.3 | 56.8 | 90.6 | 13.4 |
| *Models trained with CIDEr optimization* | | | | | | | | | | | | | | | | | | | | |
| FIBER-B | 33.3 | 30.4 | 59.9 | 102.7 | 14.1 | 29.8 | 28.9 | 58.6 | 95.3 | 13.6 | 20.7 | 25.5 | 55.1 | 83.4 | 12.3 | 28.6 | 28.5 | 58.2 | 94.1 | 13.4 |
| FIBER-GOLD-B | 34.6 | 30.9 | 60.6 | 104.7 | 14.4 | 31.3 | 29.4 | 59.4 | 98.7 | 13.9 | 21.2 | 25.9 | 55.4 | 85.7 | 12.7 | 29.9 | 29.0 | 58.8 | 97.1 | 13.8 |

**Table 7:** The complete set of results on the NoCaps test set. B@4: BLEU@4, M: METEOR, R: ROUGE-L, C: CIDEr, S: SPICE.

| Text Encoder | QQP | MNLI | QNLI | SST2 | CoLA | MRPC | STSB | RTE |
|---|---|---|---|---|---|---|---|---|
| RoBERTa-B [8] | 91.31 | 87.53 | 92.61 | 94.38 | 58.72 | 91.03 | 90.15 | 71.24 |
| METER-RoBERTa-B [5] | 91.34 | 87.38 | 92.67 | 93.92 | 57.88 | 90.57 | 89.93 | 70.28 |
| SimVLM-B [15] | 90.4 | 83.4 | 88.6 | 90.9 | 46.7 | 84.4 | - | 63.9 |
| FIBER-RoBERTa-B | 91.60 | 86.23 | 91.34 | 92.66 | 59.56 | 90.72 | 89.77 | 62.09 |

**Table 9:** Performance of text encoders on the GLUE dev sets.

where the answers to the out-of-domain questions do not appear in the top-$k$ ($k = 3,129$) candidates. We use the Karpathy split [6] in this setting.

As shown in Table 8, our generative model can perform better than VL-T5 and VL-BART, while lagging behind SimVLM especially in out-of-domain settings, possibly because SimVLM is trained with over a billion image-caption pairs and is more robust in this setting. The results indicate that our model can be turned into a general open-ended VQA model as well.

**Uni-modal Performance.** It can be interesting to see whether our backbones can still perform uni-modal tasks after VLP. Therefore, in this part, we also evaluate our language backbones on uni-modal tasks.

| Image Encoder | ImageNet | ADE20k |
|---|---|---|
| Swin-B | 86.3 | 51.6 |
| FIBER-Swin-B | 86.0 | 52.0 |

**Table 10:** Performance of image encoders on image classification and semantic segmentation.

Specifically, we test our language backbone after the first coarse-grained pre-training stage on the GLUE [13] benchmark. As shown in Table 9, the uni-modal performance of our text encoder can drop marginally on some tasks, possibly because the model only encounters simple text captions during VLP. However, it is still better than SimVLM which is trained with 800GB of web-crawled documents from scratch.

On the other hand, as shown in Table 10, our image encoder can achieve comparable and sometimes even better performance on vision-only tasks including image classification and semantic segmenta-

| Model | Shot | PascalVOC | AerialDrone | Aquarium | Rabbits | EgoHands | Mushrooms | Packages | Raccoon | Shellfish | Vehicles | Pistols | Pothole | Thermal | Avg |
|---|---|---|---|---|---|---|---|---|---|---|---|---|---|---|---|
| GLIP-B | 1 | 51.7 | 25.0 | 34.0 | 69.2 | 67.4 | 63.4. | 54.4 | 56.5 | 14.1 | 57.9 | 51.7 | 15.6 | 70.2 | 48.5 |
| GLIP-B | 3 | 57.4 | 25.2 | 44.9 | 65.1 | 69.3 | 88.3 | 67.3 | 52.8 | 28.9 | 60.7 | 62.1 | 31.3 | 67.9 | 55.5 |
| GLIP-B | 5 | 57.9 | 28.1 | 44.1 | 64.6 | 68.1 | 85.1 | 74.2 | 60.8 | 24.0 | 61.9 | 59.1 | 33.7 | 75.2 | 56.7 |
| GLIP-B | 10 | 60.6 | 27.8 | 49.7 | 67.8 | 65.2 | 87.4 | 67.5 | 54.5 | 42.3 | 65.1 | 63.9 | 39.2 | 78.7 | 59.2 |
| GLIP-B | All | 62.2 | 36.0 | 55.3 | 74.0 | 79.8 | 88.1 | 74.3 | 64.1 | 47.0 | 64.4 | 72.4 | 56.5 | 81.1 | 65.8 |
| FIBER-B | 1 | 55.7 | 25.0 | 37.9 | 69.8 | 67.2 | 83.0 | 73.2 | 54.7 | 29.7 | 58.0 | 44.8 | 27.8 | 67.0 | 53.4 |
| FIBER-B | 3 | 59.8 | 28.2 | 42.9 | 71.5 | 68.4 | 88.1 | 65.6 | 64.6 | 38.6 | 61.8 | 47.8 | 37.5 | 68.5 | 57.2 |
| FIBER-B | 5 | 61.6 | 30.9 | 49.5 | 72.3 | 69.2 | 87.5 | 73.2 | 57.4 | 38.2 | 62.7 | 55.3 | 40.3 | 61.8 | 58.4 |
| FIBER-B | 10 | 60.7 | 31.2 | 52.0 | 68.8 | 70.4 | 88.1 | 69.3 | 53.8 | 41.7 | 66.6 | 62.2 | 46.8 | 74.8 | 60.5 |
| FIBER-B | All | 68.7 | 35.4 | 58.3 | 75.9 | 79.4 | 88.1 | 72.2 | 52.9 | 45.2 | 66.1 | 72.2 | 60.9 | 80.6 | 65.9 |

**Table 11:** Few-shot and full fine-tuned results on the various ODinW datasets.

tion after coarse-grained pre-training. The results suggest that the image encoder can remain powerful even if the fusion modules are removed.

**Using the Model Checkpoint After Fine-grained Pre-training for VQA.** We also test what if we fine-tune the fine-grained pre-trained checkpoint on VQA in this part. We find that after the second-stage fine-grained pre-training, the model performance on the VQAv2 test-dev set can drop from 78.55 to 74.3, indicating that the two-stage pre-training paradigm is indeed necessary for different VL tasks, as tasks of different characteristics can require checkpoints from different pre-training stages.

**Detailed Results on ODinW.** Detailed results on the 13 ODinW datasets are provided in Table 11.

## A.4 Visualization after coarse-grained pre-training

We also provide a qualitative analysis of our model. As shown in Figure 3, we use Grad-CAM [11] to visualize the cross-attention maps of our coarse-grained pre-trained checkpoint. We find that the model can correctly align concepts and image regions for some examples, suggesting that the model can learn visual grounding implicitly.

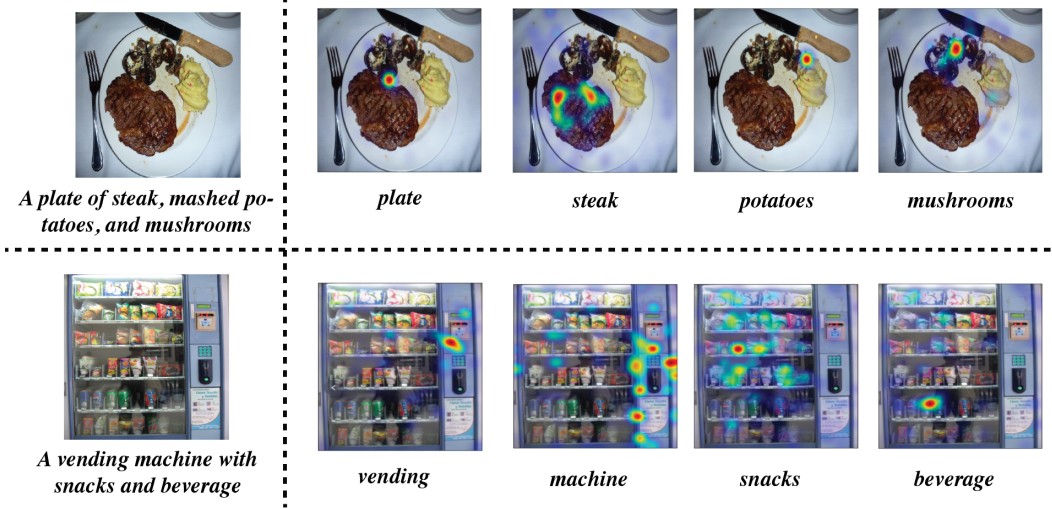

**Figure 3:** Visualizations of the cross-attention maps obtained by Grad-CAM [11]. Given each of the tokens in a caption, the model can attend to its corresponding regions. The figures are from the NoCaps validation set (ID: 253, 3766).

## A.5 Visualization after fine-grained pre-training

We also provide visualization after fine-grained pre-training in Figure 4, 5, 6, and 7.

## A.6 Text and Vision Backbones of Different Models

In this section, we list the backbones of different models in Table 12.

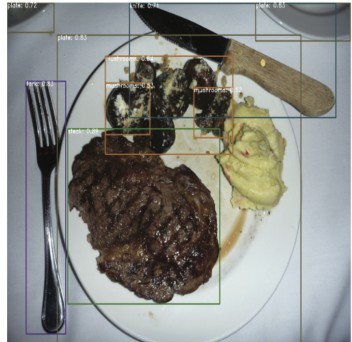 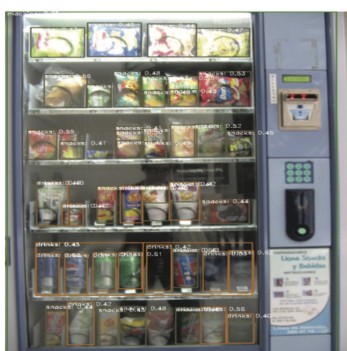

*steak. mushrooms. plate. knife. spoon. fork. mashed potatoes.*

*vending machine. drinks. snacks.*

**Figure 4:** The same images probed after fine-grained pre-training.

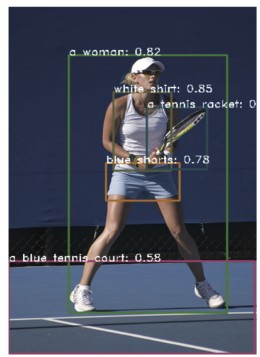 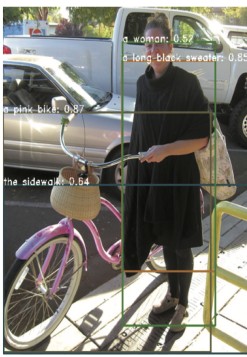 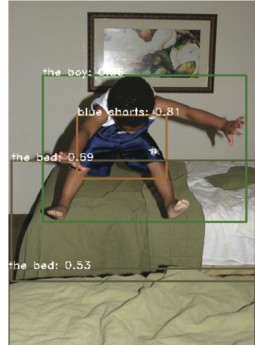

*A woman in blue shorts and white shirt holds a tennis racket on a blue tennis court.*

*A woman wearing a long black sweater is standing near a pink bike on the sidewalk.*

*The boy in blue shorts is bouncing on the bed.*

**Figure 5:** Some examples of phrase grounding from the validation set for Flickr30k entities.

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

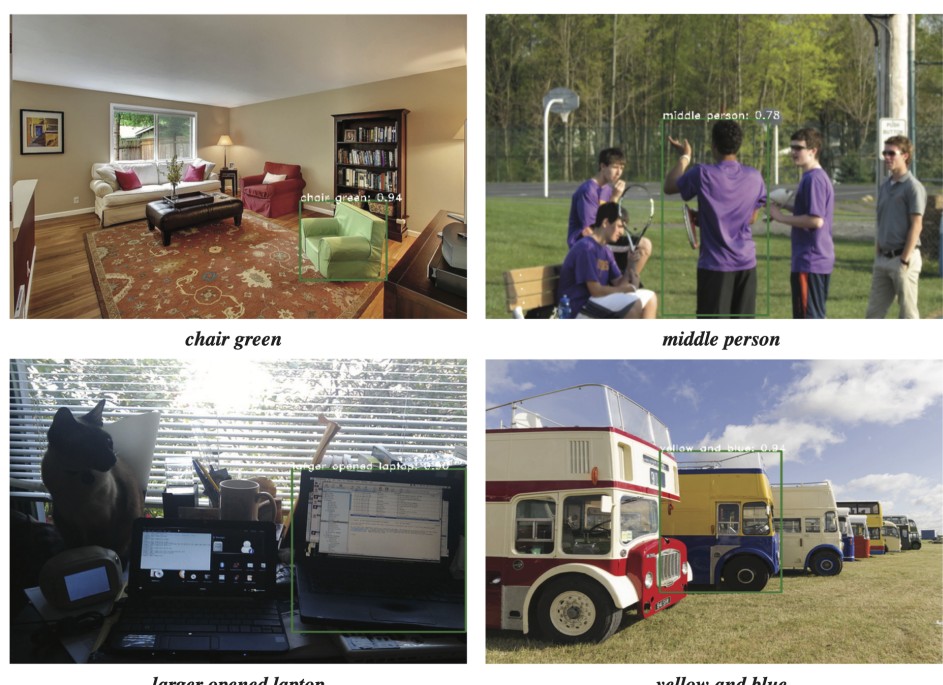

*chair green*

*middle person*

*larger opened laptop*

*yellow and blue*

**Figure 6:** Some examples of referring expression comprehension from the validation set of RefCOCO+.

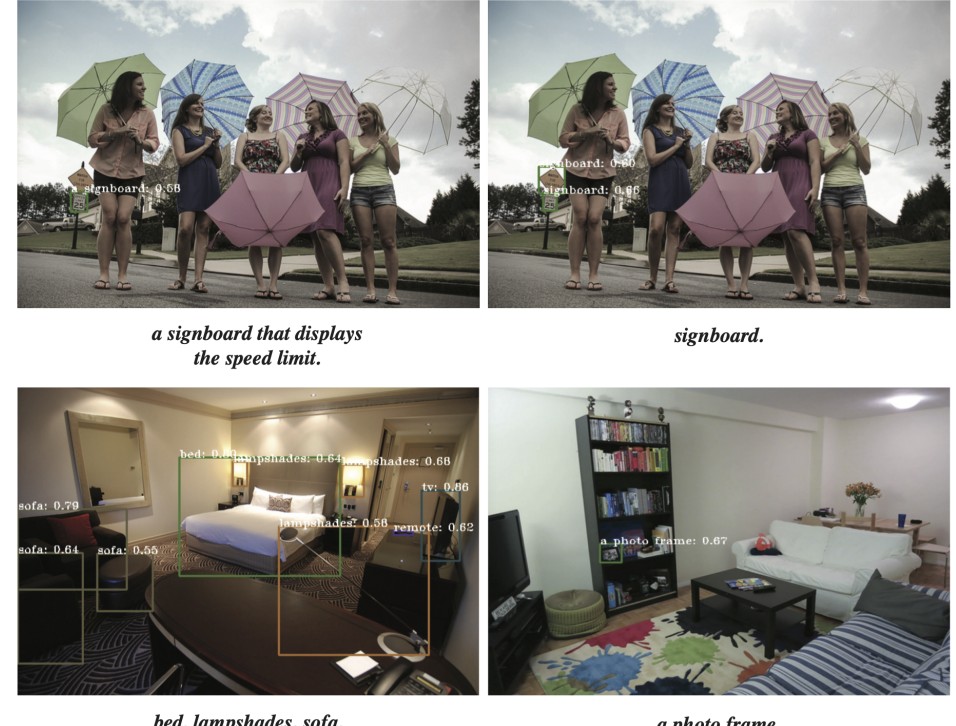

*a signboard that displays
the speed limit.*

*signboard.*

*bed. lampshades. sofa.
tv. fridge. remote.*

*a photo frame.*

**Figure 7:** Some images with prompts for various items in the scene.

| Model | Vision Encoder | Text Encoder |
|---|---|---|
| UNITER | Frozen ResNet-based OD | XFM (init w/ BERT) |
| VILLA | Frozen ResNet-based OD | XFM (init w/ BERT) |
| UNIMO | Frozen ResNet-based OD | XFM (init w/ RoBERTa) |
| VinVL | Frozen ResNet-based OD | XFM (init w/ BERT) |
| ViLT | XFM (init 5w/ ImageNet-ViT) | XFM (init w/ ImageNet-ViT and BERT embeddings) |
| ALBEF | XFM (init w/ ImageNet-ViT) | XFM (init w/ BERT) |
| VLMo | XFM (init w/ ImageNet-ViT) | XFM (init w/ ImageNet-Language-ViT) |
| UFO | XFM (init w/ ImageNet-ViT) | XFM (init w/ ImageNet-ViT) |
| ViTCAP | XFM (init w/ ImageNet-ViT) | XFM (init w/ ImageNet-ViT and BERT embeddings) |
| METER-Swin | Swin (init w/ ImageNet-Swin) | XFM (init w/ RoBERTa) |
| METER-CLIP | XFM (init w/ CLIP-ViT) | XFM (init w/ RoBERTa) |
| MDETR | EfficientNet | XFM (init w/ RoBERTa) |
| GLIP | Swin (init w/ ImageNet-Swin) | XFM (init w/ BERT) |
| FIBER | Swin (init w/ ImageNet-Swin) | XFM (init w/ RoBERTa) |

**Table 12:** Backbones of different models. XFM stands for Transformer

[10] Kishore Papineni, Salim Roukos, Todd Ward, and Wei-Jing Zhu. BLEU: a method for automatic evaluation of machine translation. In *ACL*, 2002. 1

[11] Ramprasaath R Selvaraju, Michael Cogswell, Abhishek Das, Ramakrishna Vedantam, Devi Parikh, and Dhruv Batra. Grad-CAM: Visual explanations from deep networks via gradient-based localization. In *ICCV*, 2017. 6

[12] Ramakrishna Vedantam, C Lawrence Zitnick, and Devi Parikh. CIDEr: Consensus-based image description evaluation. In *CVPR*, 2015. 1

[13] Alex Wang, Amanpreet Singh, Julian Michael, Felix Hill, Omer Levy, and Samuel Bowman. GLUE: A multi-task benchmark and analysis platform for natural language understanding. In *ICLR*, 2019. 5

[14] Wenhui Wang, Hangbo Bao, Li Dong, and Furu Wei. VLMo: Unified vision-language pre-training with mixture-of-modality-experts. *arXiv preprint*, 2021. 2

[15] Zirui Wang, Jiahui Yu, Adams Wei Yu, Zihang Dai, Yulia Tsvetkov, and Yuan Cao. SimVLM: Simple visual language model pretraining with weak supervision. In *ICLR*, 2022. 4, 5