# OpenReview forum: "Coarse-to-Fine Vision-Language Pre-training with Fusion in the Backbone "
_NeurIPS.cc/2022/Conference — NeurIPS 2022 Accept_

### Official Review · Reviewer_Gvbu · 2022-07-08

**Rating:** 5
**Confidence:** 5
**Soundness:** 3 good
**Presentation:** 3 good
**Contribution:** 2 fair

**Summary:**

This paper proposes FIBER, a multimodal transformer architecture with two sets of cross-attentions: image-to-text and text-to-image. The model is pre-trained with two stages. The first stage uses image-text pairs to learn vision-language interaction with some standard pre-training objectives. The second stage uses images with bounding box annotations to learn language-grounded localization. The pre-trained model is evaluated on a diverse set of vision-language and object detection benchmarks.

**Questions:**

- The coarse-to-fine two-stage pre-training is claimed as a key contribution. However, from the phrase grounding experiments in Table 6, it seems that coarse-grained pre-training does not help much. Does coarse-grained pre-training help the other two localization tasks, i.e., REC and OD?

- When training with object-level bounding box annotations, how does FIBER convert object names into texts? It is an important piece of information that need to be discussed in the paper.

- The paper claims that the image-text fusion mechanism from FIBER is better than that of GLIP. Could the authors provide some evidence to support this claim?

- FIBER can support training and inference with higher-resolution images due to the Swin Transformer. However, this also increases the training and inference time. It would be good if the authors could make a thorough comparison between Swin Transformer and ViT for vision-language tasks. Similarly, it is unclear how much gain does RoBERTa provide over BERT.



**Limitations:**

The authors have addressed the potential social impact. It would be also good to see some discussions on the technical limitation of the proposed method.

**Strengths And Weaknesses:**

Strengths: The paper is well-written and very easy to follow. The paper clearly states its improvement over existing methods: an additional text-to-image cross-attention mechanism and an additional stage of fine-grained pre-training. The paper does a good job to highlight the architectural differences between FIBER and existing models. The evaluation covers a sufficient number of downstream tasks.

Weaknesses: I have two major concerns about the paper as explained below.
1. The paper puts together multiple techniques to build a model. However, most of the techniques have been proven to work well by existing literature. Therefore, the insight from this paper is not clear to me. The paper does not have enough ablation experiments to analysis what effect each technique brings, and how the proposed architectural design compares with some alternative design choices. To put it short, I find it hard to learn something from this paper that can inspire future research.
2. The paper needs to be more careful when claiming better performance over existing models. The number of pre-training images is an important factor for pre-training performance, but not the only factor. FIBER uses Swin Transformer and RoBERTa, which are more powerful unimodal backbones than some of the existing ones. From the appendix, it also seems that FIBER uses higher-resolution images when finetuning on the downstream tasks, compared to most existing models.

---

> ### Author Response · Authors · 2022-08-02
> **Response (1/5)**
>
> We thank you for your valuable feedback. We are glad that you found our paper easy to follow, and that you consider our experiments on a wide variety of tasks to be sufficient to evaluate its performance. Below, we address your concerns in detail.

---

> > ### Author Response · Authors · 2022-08-02
> > **Response (2/5)**
> >
> > **Q1: The insights from this paper is not clear. I find it hard to learn something from this paper that can inspire future research.**
> >
> > We believe our work provides valuable insights to the community. Below, we summarize our insights from two perspectives: (1) coarse-to-fine pre-training, and (2) fusion in the backbone.
> >
> > For **coarse-to-fine pre-training**, our insights are:
> > * Compared with early approaches (e.g., UNITER, VinVL) that first perform supervised object detection (OD) pre-training followed by VL pre-training, we advocate a different training paradigm: first perform end-to-end VL pre-training, followed by grounded OD pre-training, so that the model can learn from large-scale image-text data and also perform language-aware OD.
> > * Compared with recent end-to-end VLP methods, which either (1) only perform coarse-grained pre-training (e.g., ALBEF, METER, BLIP, VLMo) that is suitable for captioning, VQA and retrieval, or (2) only perform fine-grained pre-training (e.g., MDETR, GLIP) that is suitable for grounding and detection, here, we propose to combine them together, allowing us to design suitable strategies for tasks of different nature.
> > * Specifically, **fine-grained pre-training** gives best results at high resolution of input images at the cost of increasing computations, which **can be avoided for image-level tasks such as captioning and VQA**.
> > * Coarse-grained training is easier to scale up and provides **an alternative cheaper approach to scaling up fine-grained pre-training** without requiring expensive pseudo-annotation like GLIP and subsequent training on image-text-box data.
> > * Fine-grained pre-training can benefit from coarse-grained pre-training when using deep multimodal fusion shared across stages.
> >
> > For **fusion in the backbone**, our insights are:
> > * In order to perform coarse-to-fine pre-training, we need a new architecture that can be shared across the two pre-training stages. Instead of having dedicated fusion layers after the uni-modal backbones, **we insert cross-attention modules into the image and text backbones, bringing gains in terms of memory and performance**.
> > * We apply this new design to Swin-base and RoBERTa, and our detailed ablations provide insights on the best-performing methods to do so (see our response to Q2 below).
> > * Using this novel architecture and pre-training strategy, we are able to obtain **SOTA performance on a wide range of VL tasks as well as on OD**. We would like to underline that this is not trivial, and there is no existing work that has performed all the tasks within one framework.
> >
> > Leveraging these two insights, our proposed architecture coupled with our novel two-stage pre-training pipeline achieves better performance than models using magnitudes more data on coarse-grained tasks (Table 3 & Line 263-265) and models using up to 25x more box-annotated data on fine-grained tasks (Table 5 & Line 281-286).
> >
> > **Q2: The paper does not have enough ablations.**
> >
> > Below, we summarize the ablations present in Appendix A.2 & A.3 that we hope will provide useful insights to readers.
> > * Co-attention works similarly to merged attention for fusion in the backbone.
> > * *Adding a gating parameter in co-attention* allows adding fusion in more layers, and also gives better performance than merged attention.
> > * Adding co-attention in the last 6 layers balances well between efficacy and efficiency.
> > * *MLM, ITM with hard negative mining, and ITC* are all important pre-training objectives for training FIBER-style models.
> > * Empirically, we show better performance compared to other models using the similar image and text backbones, such as METER [2] and GLIP [3], and models with other types of backbones, on various tasks.
> > * Additionally, in terms of efficiency, *our fusion in the backbone model consumes half of the FLOPs needed by models using the similar image and text backbones* with late fusion such as METER [2] (line 152, 12.35 vs. 24.04 GFLOPs for one instance), and when compared in terms of training speed, FIBER takes 1.38 s/iteration whereas GLIP [3] takes 2.14 s/iteration (Table 2).
> >
> > For more details, please see Appendix A.2 & A.3, where we have provided detailed ablations that guided our architecture design, including ablations on fusion strategies, pre-training objectives, architecture for captioning, and additional results on open-ended VQA, image-text retrieval with re-ranking, detailed few-shot ODinW results. Due to the space limit, these ablations and additional results were only added in the Appendix.
> >
> > We hope this alleviates your concern regarding ablations to analyze what effect each technique brings. We believe that our architecture is novel, and the knowledge of how to effectively and efficiently perform multimodal fusion for such new models is a useful insight and provides a solid basis on which to build better models for future research.

---

> > > ### Author Response · Authors · 2022-08-02
> > > **Response (3/5)**
> > >
> > > **Q3: Comparisons to existing models using different image and text encoders as well as image resolution and pre-training data.**
> > >
> > > In our result tables, we ensure that we report relevant prior work displaying the current best performance on the tasks while also covering a variety of architectures using different image and text encoders. However, we also make sure to compare to the most similar prior work such as METER [2] & GLIP [3] that use the same image encoder, and similar text encoder.
> > >
> > > Further, since previous work (METER) has thoroughly analysed the effect of different text and image enoders as well as image resolutions, we build upon their work by using the suggested settings for image resolutions. Specifically, for VQA, image captioning and retrieval tasks, we follow the settings used by METER a SOTA coarse-grained model); for phrase grounding, referring expression, and OD tasks, we follow GLIP [3] (a SOTA fine-grained model), both published at CVPR 2022.
> > >
> > >
> > > METER [2] has made detailed comparisons between different vision and text backbones for VL tasks in their paper (Table 2-4), demonstrating that:
> > > * For vision encoder, CLIP-ViT-Base (resolution: 224x224) outperforms Swin Transformer-Base (resolution: 384x384) after VLP, while underperforms Swin Transformer-Base before VLP;
> > > * For text encoder, RoBERTa-Base and BERT-Base perform similarly.
> > >
> > > In their final settings,
> > > * METER [2] employs CLIP-ViT-Base or Swin Transformer as the vision encoder and RoBERTa as the text encoder;
> > > * GLIP [3] employs Swin Transformer and BERT for vision and text encoders.
> > >
> > > Since we are interested in building a unified model that can solve both coarse and fine-grained tasks, we choose to use RoBERTa as the text encoder, and Swin Transformer as the image encoder, because the encoded hierarchical multi-scale image features are more naturally suitable for object detection tasks. It is important to note that even though METER proved that the CLIP-ViT-Base is better than Swin-Base for VL tasks after VLP (see Table 8 in METER), *our improved fusion in the backbone architecture using a Swin is able to outperform METER [2] on all the image-level tasks, and also outperform GLIP on fine-grained tasks while keeping pre-training data fixed.*
> > >
> > > In summary, we believe we have made fair comparisons with the strongest baselines in the literature.
> > >
> > > **Q4: From the phrase grounding experiments in Table 5, it seems that coarse-grained pre-training does not help much.**
> > >
> > > Though the FIBER results with and without coarse-grained pre-training do not differ much on R@5 and R@10 scores, we would like to emphasize that **for the more challenging R@1 score, FIBER can improve it from 86.5 to 87.4 with coarse-grained pre-training**, which we consider to be significant. If we look at Table 5 more closely, **our FIBER-B model even outperforms GLIP-L trained on 27M images with box level annotations** (87.4 vs. 87.1 on R@1 score), which is achieved solely due to the coarse-grained pre-training. This achievement highlights the fact that it is not necessary to pseudolabel a huge corpus (27M as in GLIP) to do well on phrase grounding tasks, but instead, our proposed coarse-to-fine grained pre-training strategy can be used as an alternative, and we believe this is a very useful result, and hence feel it is fair to claim the coarse-to-fine pre-training as a key contribution.
> > >
> > > **Q5: On the other two localization tasks (REC, OD), does coarse-grained pre-training help?**
> > >
> > > Our paper proposes a new architecture as well as the coarse-to-fine pre-training strategy. Our architectural improvements already bring us considerable gains compared to prior work on a large variety of tasks, including fine-grained tasks such as OD, REC and phrase grounding. By using the coarse-to-fine strategy, we further push the results up by a few points. On datasets such as RefCOCO+ and Flickr30k where the performance is already so close to the ceiling, we believe that achieving a few points is still meaningful as the numbers are already very saturated. We ran additional experiments to answer your question about how well FIBER does without C.G. pre-training and summarize the results in the table below. We see gains across both tasks (REC & OD) when utilizing the coarse-grained pre-training. Similar to the case of Flickr30k, **on RefCOCO+ the coarse-grained pre-training helps FIBER to get better performance than Large sized model trained with more data**.
> > >
> > > | Model | OD on COCO | OD on LVIS | ODinW | RefCOCO+|
> > > | -------- | -------- | -------- | -------- | -------- |
> > > |      | Zero-shot/Finetune     | Zero-shot/Finetune    | Zero-shot  | Val/TestA/TestB|
> > > | OFA-L    | -    |    -    |    - |    84.49/90.10/ 77.77 |
> > > | GLIP-B     | 48.1/57.0     | 29.1/51.0     | 44.8     | -    |
> > > | FIBER-B w/o C.G. VLP     | 48.9/57.8     | 31.6/55.8     | 45.1     | 85.04/88.82/78.59|
> > > | FIBER-B     | 49.3/58.4     | 35.8/56.9     | 47.0     | 85.74/90.13/79.38|

---

> > > > ### Author Response · Authors · 2022-08-02
> > > > **Response (4/5)**
> > > >
> > > >
> > > > **Q6: When training with object-level bounding box annotations, how does FIBER convert object names into texts?**
> > > >
> > > > We apologize for the lack of clarity on this matter. The object category names are directly used in their text form separated by full stops as input to the text encoder. We follow the same protocol as in GLIP [3] to be comparable to their experiments. More specifically, the input text will look like this:  "person. bicycle. car. .... toothbrush", and the model will learn how to ground image regions into these object names. This effectively makes our FIBER model suitable to object detection tasks. An example of input and output predicted by the model can be seen in Fig. 12 of the Appendix.
> > > >
> > > >
> > > > **Q7: The paper claims that the image-text fusion mechanism from FIBER is better than that of GLIP. Could the authors provide some evidence to support this claim?**
> > > >
> > > > We prove this from the following perspectives.
> > > > * In terms of performance on region-level tasks, by comparing the results of Row #1 and #2 in the table below, it is clear that the fusion mechanism of FIBER is better than that of GLIP.
> > > > * In terms of efficiency, as summarized in Table 2 of the paper, FIBER takes 1.38 s/iteration whereas GLIP takes 2.14 s/iteration under a strictly fair comparison.
> > > >
> > > >
> > > > | | Model | OD on COCO | OD on LVIS | ODinW |
> > > > | -------- | -------- | -------- | -------- | -------- |
> > > > | |      | Zero-shot/Finetune     | Zero-shot/Finetune    | Zero-shot  |
> > > > | 1| GLIP-B     | 48.1/57.0     | 29.1/51.0     | 44.8     |
> > > > | 2| FIBER-B w/o C.G. VLP     | 48.9/57.8     | 31.6/55.8     | 45.1     |
> > > > | 3| FIBER-B     | 49.3/58.4     | 35.8/56.9     | 47.0     |
> > > >
> > > > Further, we would like to stress here on the fact that **GLIP [3] cannot be used for VL tasks that require deep fusion such as VQA and captioning.** Even compared to a newer version of the paper (GLIPv2 [4] on arxiv on 6/12/2022), we still **outperform their approach (GLIPv2-Base) on VQA by 5 points (78.46 vs 73.3)**, even though they use much more data (20M compared to our 4M), and have to train at high resolution (1333x800), even for tasks such as VQA, clearly displaying the efficacy of FIBER and the two-stage pre-training strategy that we propose. During inference, due to its use of OD to first extract region features, GLIPv2 [4] will be also much slower than FIBER on VQA and image captioning tasks.
> > > >
> > > >
> > > > **Q8: FIBER can support training and inference with higher-resolution images due to the Swin Transformer. However, this also increases the training and inference time. It would be good if the authors could make a thorough comparison between Swin Transformer and ViT for vision-language tasks.**
> > > >
> > > > ViT-based models are typically constrained to use images at low resolution, due to their quadratic complexity in terms of the size of the image. For object detection (OD) and other region-level tasks that require high-resolution inputs, this becomes a computational challenge, while the Swin transformer allows the computational complexity to remain linear in the size of the image. Furthermore, the encoded multi-scale image features from Swin also make it more suitable for OD tasks. We choose to use Swin for this reason, even though METER [2] proves that it achieves worse performance than CLIP-ViT on image-level tasks after VLP (Please see the response to Q3).
> > > >
> > > > METER [2] has also compared the computational time between Swin Transformer and ViT for VL tasks (in their Table 12), showing that Swin Transformer (resolution 384x384) has a similar inference time with ViT (resolution 224x224). This means that image resolution is not the sole factor for inference time. Further, we show (Table 2 in the paper) that even at higher resolutions, we are saving in terms of training time as well as parameters compared to models using similar backbones (such as GLIP [3]).
> > > >
> > > >
> > > > To further address your concern, we have also conducted additional experiments comparing Swin and ViT in the table below, listed under Q9.
> > > >
> > > > **Q9: Similarly, it is unclear how much gain does RoBERTa provide over BERT.**
> > > >
> > > > METER [2] has conducted an extensive survey of different backbones, and they note that RoBERTa-Base and BERT-Base perform similarly both with and without VLP.
> > > >
> > > >
> > > > We also conducted additional experiments in our settings without VLP, which has been included in our revision.
> > > >
> > > > | Model | Image Encoder | Text Encoder | VQAv2 test-dev |
> > > > | -------- | -------- | -------- | -------- |
> > > > | FIBER  | Swin | RoBERTa | 71.97 |
> > > > | FIBER  | Swin | BERT | 71.86 |
> > > > | FIBER  | CLIP-ViT | RoBERTa | 71.37 |
> > > >
> > > > The results confirmed the findings of METER [2] that RoBERTa and BERT perform similarly and we find that even with fusion in the backbone, Swin performs slightly better than CLIP-ViT before VLP.

---

> > > > > ### Author Response · Authors · 2022-08-02
> > > > > **Response (5/5)**
> > > > >
> > > > >
> > > > > **Q10: It would be also good to see some discussions on the technical limitation of the proposed method.**
> > > > >
> > > > > Thank you for the suggestion! While our proposed framework has been evaluated on a wide range of tasks, it has not been extended to tasks such as semantic/instance/panoptic segmentation yet. In addition, our model relies on pre-trained image and text encoders, thus it would be worth investigating if we can train the whole model from scratch. We leave these as our future work.
> > > > >
> > > > > [2] Dou, Zi-Yi, et al. "An empirical study of training end-to-end vision-and-language transformers." CVPR 2022.
> > > > >
> > > > > [3] Li, Liunian Harold, et al. "Grounded language-image pre-training." CVPR 2022.
> > > > >
> > > > > [4] Zhang, Haotian et al. “GLIPv2: Unifying Localization and Vision-Language Understanding.” ArXiv abs/2206.05836 (2022)

---

> ### Author Response · Authors · 2022-08-05
> **Reply**
>
> Dear Reviewer,
>
> We are truly thankful for your valuable feedback! We have tried to address all of your concerns in our responses. As the author-reviewer discussion period will end soon (until Aug. 9), we would love to hear if you still have any concerns and we are more than happy to discuss.

---

> ### Author Response · Authors · 2022-08-08
> **Looking forward to your feedback**
>
> Dear reviewer, as the discussion period will end in 24 hours, we would love to hear if you still have any concerns and we are willing to discuss them. Thank you for your time!

---

> > ### Comment · Reviewer_Gvbu · 2022-08-09
> > **Thanks for the response**
> >
> > I appreciate the authors comprehensive response and new results. While I still find the two-stage training to have marginal improvements on some tasks, it is good to see more insights and the improvement over GLIP. Therefore, I would raise my score by 1.

---

> > > ### Author Response · Authors · 2022-08-09
> > > **Thank you**
> > >
> > > Thank you for the update!
> > >
> > > We would like to kindly point out that we have evaluation results on ~30 datasets in the paper. Many of these tasks (e.g. VQAv2, COCO object detection, RefCOCO+) are highly competitive, yet we can obtain consistent performance improvements over strong baselines, often outperforming methods using magnitudes more data, which we believe is a significant achievement.

---

### Official Review · Reviewer_fMci · 2022-07-10

**Rating:** 7
**Confidence:** 4
**Soundness:** 2 fair
**Presentation:** 2 fair
**Contribution:** 3 good

**Summary:**

This paper proposes to fuse visual and textual features with cross attention in the backbones for vision-language pre-training, which makes the model applicable to both high-level vision-language understanding tasks (e.g., VQA, captioning) and region-level image understanding tasks (e.g., object detection, phrase grounding). Moreover, the model is pre-trained in two-stage coarse-to-fine manner (i.e., first high-level VL tasks then region-level VL tasks). Experiments are conducted on a wide range of VL tasks to demonstrate the superiority of the proposed method.

**Questions:**

Pls check the weakness section.

**Limitations:**

The pre-trained model in this paper can not be directly transferred to all the considering VL tasks in zero-shot manner.

**Strengths And Weaknesses:**

Strengths

(1)	The proposed model is the first VLP model that can be applicable to both high-level and region-level downstream VL tasks.

(2)	A two-stage pre-training strategy is proposed to pre-train the model in coarse-to-fine manner.

(3)	Superior performances are achieved compared with SOTA methods on a wide range of VL tasks.

Weaknesses

(1)	Some related works that jointly support VQA, Retrieval, and captioning are missing:
[A] Unified vision-language pre-training for image captioning and vqa[C]//Proceedings of the AAAI Conference on Artificial Intelligence. 2020, 34(07): 13041-13049.
[B] Scheduled sampling in vision-language pretraining with decoupled encoder-decoder network[C]//Proceedings of the AAAI Conference on Artificial Intelligence. 2021, 35(10): 8518-8526.

(2)	As shown in Table 5, it is better to indicate the type of vision/text backbone of each run in other tables (Table 3, 4, 6, 7), pursuing a fair comparison under same vision/text backbone.

(3)	The ablation study about the effect of the number of layers to additionally fuse cross-modal features in the backbones is missing.

---

> ### Author Response · Authors · 2022-08-02
> **Response**
>
> **Related work:**
>
> Thank you for the reference! We have included these papers in the updated version, and we would just like to mention that while building VL models that jointly support image-level tasks (e.g., VQA, retrieval, and captioning) has become a standard practice in end-to-end VLP (see paragraph *VLP for Classical VL Tasks* in Sec. 2), we focus on a unified framework for both image-level tasks and region-level tasks such as phrase grounding, object detection, and referring expression comprehension.
>
> **Indicating text/vision backbones of different models:**
>
> Thank you for the suggestion! Some models make special modifications to the backbones and have no clear boundaries between vision and text backbones, hence including all the vision/text backbones would have made the tables too crowded, and we decided to not add it into the main paper submission. The details of these models are provided below, which has also been included in Appendix in our revision. OD is short for Object Detection.
>
>
> | Model | Image Backbone | Text Backbone |
> | -------- | -------- | -------- |
> | UNITER | Frozen ResNet-based OD | Transformer (initialized w/ BERT) |
> | VILLA | Frozen ResNet-based OD | Transformer (initialized w/ BERT) |
> | UNIMO | Frozen ResNet-based OD |  Transformer (initialized w/ RoBERTa)|
> | VinVL | Frozen ResNet-based OD | Transformer (initialized w/ BERT) |
> |ViLT | Transformer (initialized w/ ImageNet-ViT) | Transformer (top layers initialized w/ ImageNet-ViT and the embedding layer initialized with BERT) |
> | ALBEF | Transformer (initialized w/ ImageNet-ViT)  | Transformer (initialized w/ BERT) |
> |VLMo |  Transformer w/ MoE (initialized w/ ImageNet-ViT) | Transformer w/ MoE (initialized w/ ImageNet-ViT further pre-trained on language data) |
> | UFO | Transformer (initialized w/ ImageNet-ViT) | Transformer (initialized w/ ImageNet-ViT) |
>  |ViTCAP | Transformer (initialized w/ ImageNet-ViT) | Transformer (top layers initialized w/ ImageNet-ViT and the embedding layer initialized with BERT) |
> |METER-Swin | Swin (initialized w/ ImageNet-Swin) | Transformer (initialized w/ RoBERTa)|
> | METER-CLIP | Transformer (initialized w/ CLIP-ViT) | Transformer (initialized w/ RoBERTa) |
> | MDETR | EfficientNet| Transformer (initialized w/ RoBERTa) |
> | GLIP | Swin (initialized w/ ImageNet-Swin) | Transformer (initialized w/ BERT) |
> | FIBER | Swin (initialized w/ ImageNet-Swin) | Transformer (initialized w/ RoBERTa)  |
>
>
> **The effect of the number of layers to fuse cross-modal features:**
>
> Thank you for pointing this out! We do have this ablation, as well as several other ablations in Appendix A.2. Due to the page limit and the number of downstream tasks that we have covered in this paper, we present the main evaluation results in the main content and put the ablations and analyses in the appendix. For your convenience, we have also included this ablation study below.
>
> | #Fusion Layers | #Fusion Params. (M) | VQAv2 | $\Delta$ |
> | -------- | -------- | -------- | -------- |
> | 0 | 0 | 65.65 | --- |
> | 3 | 16.1 | 71.20 | +5.5 |
> | 6 | 26.0 | 71.97 | **+0.77** |
> | 9 | 35.8 | 72.10 | +0.13 |
> | 12 | 45.6 | 72.08 | -0.02|
>
> As seen here, the performance gap when going from 3 layers to 6 layers is much larger than the smaller gaps between 6 to 9 and 12 layers. We choose to use 6 layers as a good tradeoff between performance and parameter efficiency.

---

> > ### Comment · Reviewer_fMci · 2022-08-08
> > **Thanks for your reply**
> >
> > I have gone through the comments from the other reviewers and the responses from the authors, and most of my concerns have been addressed well. The proposed two-stage training strategy and the architecture are novel enough. Also, a comprehensive analysis and ablation study are included in the paper. Thus, my final recommendation for this paper is Accept.

---

> > > ### Author Response · Authors · 2022-08-08
> > > **Thank you**
> > >
> > > Thank you for the update!

---

> ### Author Response · Authors · 2022-08-05
> **Reply**
>
> Dear Reviewer,
>
> We are truly thankful for your valuable feedback! We have tried to address all of your concerns in our responses. As the author-reviewer discussion period will end soon (until Aug. 9), we would love to hear if you still have any concerns and we are more than happy to discuss.

---

### Official Review · Reviewer_SULZ · 2022-07-11

**Rating:** 6
**Confidence:** 4
**Soundness:** 3 good
**Presentation:** 3 good
**Contribution:** 3 good

**Summary:**

This paper proposes a new vision-language model that can deal with both  VL tasks and region-level understanding tasks.
The proposed model, Fiber inserts cross-attention to the model to learn multimodal fusion.
The pretraining leverages two kinds of data: coarse-grained image-text pairs and fine-grained image-text-box data.


**Questions:**

Why do not compare the performance with X-VLM, which is a recent vision-language pre-trained model?

**Strengths And Weaknesses:**

Strengths:
	1. The model is novel. The proposed fusion in the backbone is a good architecture by inserting cross-attention layers. I like this solution which is more light-weight than previous fusion modules. The authors demonstrate this in terms of parameter size (Fiber adds 26M parameters while METER adds 110M).
	2. Performance is good. It outperforms METER by around 2/% on VQA and NLVR. Also, it has a good performance on region-level understanding tasks.

Weaknesses:
	1. My biggest concern is a recent work X-VLM [1]. Could the authors distinguish the proposed method and X-VLM model? The motivation to leverage both coarse- and fine-grained features is similar. And the authors did not discuss the differences in the model architecture and performance. Especially, X-VLM is a one-stage model and I feel like the one-stage model is easier to implement and use.


[1] X-VLM: Multi-Grained Vision Language Pre-Training (ICML 2022)

---

> ### Author Response · Authors · 2022-08-02
> **Response (1/2)**
>
> Thank you for your feedback on our work, and for acknowledging FIBER to be a good architecture that provides a light-weight solution for multi-modal fusion. We apologize for missing the related work X-VLM [1], we agree that it is quite relevant to the discussion in our paper, and have included it in our revision.
>
> We will now address your question regarding the comparison to X-VLM along three axes - motivation of the approach, technical details, and performance on downstream tasks.
>
> **1. Motivation**
>
> | X-VLM                                                                                                                          | FIBER |
> |:------------------------------------------------------------------------------------------------------------------------------ | ----- |
> | Emphasis lies in leveraging fine-grained alignment information for image-level VL tasks such as VQA and image-text retrieval, without considering tasks such as object detection. | Emphasis lies in building a unified framework for efficient VL pre-training that benefits both image-level and region-level VL tasks, while minimizing the burden for tasks that do not require region-level pre-training. More concretely, in our approach, image-level tasks, such as VQA, image captioning and retrieval, do not make use of region-level training, as we believe they can be tackled satisfactorily at the "coarse"-grained level after our first-stage pre-training. As mentioned in line 41-49, this division of labor allows us to design strategies suitable for different kinds of tasks. For example, region-based pre-training is beneficial for tasks such as object detection and phrase grounding, giving the best results at high resolution of input images at the cost of increased computational burden, which can be avoided for tasks such as captioning, VQA and retrieval.  |
>
> **2. Technique**
>
> In terms of methodology, X-VLM has a one-stage pre-training recipe, where it mainly incorporates object- and region-level features into input representations, and proposes to add a bounding box prediction objective, while we propose a two-stage coarse-to-fine pre-training strategy and a new fusion-in-the-backbone architecture that can potentially improve methods such as X-VLM.
>
>
> | X-VLM | FIBER|
> |:-------- |:-------- |
> |**Fusion on top of the backbones**: X-VLM requires each fusion layer to be equipped with a self-attention block, a cross-attention block and a feedforward network block. | **Fusion in the backbone**: We propose a new fusion-in-the-backbone architecture that can perform modality fusion efficiently (as discussed in Line 57-70 and Sec. 3.1). Specifically, we only need to insert a single cross-attention block into each fusion layer. We have also shown concrete comparisons of the fusion in terms of number of parameters (110M added parameters for GLIP [3] and METER [2] vs 26M in FIBER) and training time (1.38 s/iteration for FIBER vs 2.14 s/iteration for GLIP [3]) of FIBER compared to METER [2] and GLIP [3], which use a later fusion strategy than our approach. In terms of FLOPS, FIBER only consumes half of the FLOPs needed by METER [2] (12.35 vs. 24.04 GFLOPs for one instance. Even with the lighter-weight approach, we are able to outperform METER [2], GLIP [3] as well as X-VLM across a wide variety of tasks |
> | **One-stage pre-training**:  X-VLM operates in a one-stage pre-training setup by using both image-level and region-level annotations simultaneously. They overcome the fact that not all datapoints have region-level annotations by ensuring half of the images in a batch contain bounding box annotations during data sampling. In cases where the coarse-grained data is scaled up, this can become a hurdle with severe data imbalance. Further, the benefits of using different resolution of inputs (especially high resolution for fine-grained tasks) is not explored. | **Two-stage pre-training**: As discussed in the paper (Line 11-15, 31-49, 71-91), we believe that separating VLP into stages can have several benefits. First, coarse-grained pre-training is easier to scale up because it is possible to crawl massive image-caption data from the Internet, while it can be costly to annotate data with bounding box information at a large scale. Second, for object detection and phrase grounding, the regions of interest can be very small, and in these cases it is favorable to train with high input resolution such as 800 x 1,333. This high resolution of input images, while ensuring better performance, makes training expensive if done at the scale of internet scraped data. This is why we propose to split pre-training into two stages to obtain the best of both worlds - ease of scaling up to obtain good performance on a variety of tasks while remaining feasible in terms of computational complexity.|

---

> > ### Author Response · Authors · 2022-08-02
> > **Response (2/2)**
> >
> >
> > **3. Downstream task performance**
> >
> > a)  Image-level tasks: Without using any region-level annotations, we pre-train our model only on data having image-caption pairs and then finetune this model on a variety of tasks. Even though we use less dense annotations, we are able to outperform X-VLM on most tasks, and we attribute this to our improved architecture.
> >
> >
> > |   Model    | VQA2  | NLVR2 | Flickr30k IR/TR | COCO IR/TR | COCO captioning w/ Cider Optimization |
> > |:----------:|:-----:|:-----:|:---------------:|:----------:|:-------------------------------------:|
> > | X-VLM (4M) | 78.09 | 84.21 |    86.1/96.8    | 63.1/80.4  |                 140.8                 |
> > | FIBER (4M) | 78.46 | 85.52 |    91.0/96.0    | 69.6/80.1  |                 142.8                 |
> >
> > b) Region-level tasks: With our second-stage pre-training bringing in fine-grained annotations, our model outperforms X-VLM on region-level tasks such as referring expression comprehension, notably by more than 2 points on the challenging testB split of RefCOCO+. While our approach can also be used seamlessly without any modifications for the task of phrase grounding on Flickr30k dataset (where each text may correspond to multiple boxes), as well as Object Detection, it is unclear how one would use X-VLM to do object detection. We report results on multiple object detection datasets, outperforming GLIP [3] which is a recent state-of-the-art model focusing mainly on detection. We believe that having a unified model that can tackle core vision tasks such as OD while also achieving state of the art on VL tasks is non-trivial.
> >
> >
> >
> > | Model |   RefCOCO+ (val/testA/testB) | COCO Val2017 (ZS/Finetune) | Flickr30k Test (R@1,5,10) |
> > |--|--|--|--|
> > |X-VLM (16M) | 84.51/89.00/76.91 | - | - |
> > |FIBER (4M+0.8M) | 85.76/90.13/79.38 | 49.3/58.4 | 87.4/96.4/97.6 |
> >
> >
> >
> >
> > We would like to conclude this comparison by noting that while very related, our approach is significantly different from X-VLM in the aforementioned ways, and we believe that our proposed fusion-in-the-backbone architecture could even complement X-VLM-like models to achieve better performance.
> >
> > [1] Zeng, Yan, Xinsong Zhang, and Hang Li. "Multi-Grained Vision Language Pre-Training: Aligning Texts with Visual Concepts." ICML 2022.
> >
> > [2] Dou, Zi-Yi, et al. "An empirical study of training end-to-end vision-and-language transformers." CVPR 2022.
> >
> > [3] Li, Liunian Harold, et al. "Grounded language-image pre-training." CVPR 2022.

---

> ### Author Response · Authors · 2022-08-05
> **Reply**
>
> Dear Reviewer,
>
> We are truly thankful for your valuable feedback! We have tried to address all of your concerns in our responses. As the author-reviewer discussion period will end soon (until Aug. 9), we would love to hear if you still have any concerns and we are more than happy to discuss.

---

### Author Response · Authors · 2022-08-02
**Joint response**

We thank the reviewers for their valuable feedback. We are encouraged that they found our work to be novel and that it provides a new architecture that is more lightweight than previous fusion modules (Reviewer SULZ). We are glad that they appreciate the performance improvements over SOTA on a wide variety of both high-level and region-level downstream VL tasks (Reviewer SULZ, Reviewer fMci, Reviewer Gvbu). We are especially pleased that (Reviewer Gvbu) found our paper to be well-written and very easy to follow while doing a good job of highlighting the architectural differences between FIBER and existing models. We address specific reviewer comments below and will incorporate all feedback into the paper. At this stage, we have added the extra details into the appendix of the revision and will reorganize it into the extra page for the camera ready. Finally, we will be releasing our code and weights to ensure reproducibility.

---

### Meta-Review · Area_Chair_TxfJ · 2022-08-26

**Recommendation:** Accept
**Confidence:** Certain

**Metareview:**

This paper proposes a two-stage pretrain visual language model which can deal with both high level and region level downstream tasks.  Experiments show significant improvement SotA models.  Main concerns from reviews are some missing references while the author gave detailed comparisons in the responses.  Although reviewer Gvbu's opinion is still somewhat conservative, I think the novelty of the paper is clear and the comparison to the SotA is sufficient.

**Award:**

No

---

### Decision · Program_Chairs · 2022-09-14

Accept